# Nucleotide exchange is sufficient for Hsp90 functions in vivo

Michael Reidy ⬡[1] ✉, Kevin Garzillo ⬡[1,2] & Daniel C. Masison[1]

Hsp90 is an essential eukaryotic chaperone that regulates the activity of many client proteins. Current models of Hsp90 function, which include many conformational rearrangements, specify a requirement of ATP hydrolysis. Here we confirm earlier findings that the Hsp82-E33A mutant, which binds ATP but does not hydrolyze it, supports viability of *S. cerevisiae*, although it displays conditional phenotypes. We find binding of ATP to Hsp82-E33A induces the conformational dynamics needed for Hsp90 function. Hsp90 orthologs with the analogous EA mutation from several eukaryotic species, including humans and disease organisms, support viability of both *S. cerevisiae* and *Sz. pombe*. We identify second-site suppressors of EA that rescue its conditional defects and allow EA versions of all Hsp90 orthologs tested to support nearly normal growth of both organisms, without restoring ATP hydrolysis. Thus, the requirement of ATP for Hsp90 to maintain viability of evolutionarily distant eukaryotic organisms does not appear to depend on energy from ATP hydrolysis. Our findings support earlier suggestions that exchange of ATP for ADP is critical for Hsp90 function. ATP hydrolysis is not necessary for this exchange but provides an important control point in the cycle responsive to regulation by co-chaperones.

Hsp90 is an abundant, highly conserved molecular chaperone that is essential in eukaryotes. It regulates a wide range of client proteins. Hsp90 is a drug target for anticancer therapies and is also involved in other human diseases such as tauopathies[1–4]. Hsp90 has an amino-terminal (N) domain that binds nucleotide, a charged linker region, a middle (M) domain that binds clients and a carboxy-terminal (C) dimerization domain. Hsp90 undergoes radical conformational shifts, largely influenced by the nucleotide state of the N-termini. In the apo or ADP-bound state, Hsp90 is in a V-shaped open conformation, with the N-termini of each protomer separated[5]. Hsp90 has higher affinity for ADP than ATP, so it might be expected that cytosolic Hsp90 mostly populates an open conformation in vivo[6]. When purified Hsp90 binds to non-hydrolysable ATP analogs such as adenylyl-imidodiphosphate (AMP-PNP) or adenosine-5′-O-3-thiophosphate (ATP-γ-S), a lid structure in the N domain repositions over the bound nucleotide. Almost simultaneously, the N domains dock onto the M domains, and the two N domains bind each other[7,8]. This so-called closed-clamp conformation of Hsp90 is much more compact and competent to hydrolyze ATP[9]. Thus, binding of non-hydrolysable ATP analogs trap Hsp90 into a pre-hydrolysis conformation[10]. It is thought that in vivo, the rearrangements that are necessary to achieve the pre-hydrolysis conformation after ATP binding are rate limiting[11,12]. This idea is supported by observations that in vitro ATP does not induce conformational rearrangements like non-hydrolysable ATP analogs[6,13–16]. In vivo, these rearrangements are regulated by co-chaperones and post-translational modifications (reviewed in:[17,18]). According to the current model of Hsp90 function, upon ATP hydrolysis the Hsp90 dimer returns to the open conformation, releasing the client, ADP and phosphate, thus resetting the reaction cycle[19–22]. However, it was reported that dimer opening is only weakly coupled to hydrolysis[15], and more recent evidence for a compact, ADP-bound conformation[16] suggests that hydrolysis is not required for reopening the Hsp90 dimer.

[1]Laboratory of Biochemistry and Genetics, National Institute of Diabetes and Digestive and Kidney Diseases, National Institutes of Health, Bethesda, MD 20892-0830, USA. [2]Present address: Department of Biological Sciences, Lehigh University, Bethlehem, PA 18015, USA. ✉e-mail: michael.reidy@nih.gov

**Table 1 | Strains used in this study**

| Strain | Species | Genotype | Ref |
|---|---|---|---|
| MR1075 | *S. cerevisiae* | Mat α; *his3Δ1; leu2Δ0; lys2Δ0; trp1Δ63; ura3Δ0; hsc82ΔHphMX; hsp82ΔNatMX*; pMR62 (P$_{HSC82}$::*HSP82/URA3*) | 33 |
| MR1112 | *S. cerevisiae* | Mat α; *his3Δ1; leu2Δ0; trp1Δ63; ura3Δ0; hsc82ΔHphMX; hsp82ΔNatMX; slt2ΔKanMX*; pMR62 (P$_{HSC82}$::*HSP82/URA3*) | This study |
| MR1123 | *S. cerevisiae* | Mat α; *his3Δ1; leu2Δ0; trp1Δ63; ura3Δ0; hsc82ΔHphMX; hsp82ΔNatMX; PKC1-GFP*::His3MX; pMR62 (P$_{HSC82}$::*HSP82/URA3*) | This study |
| MRP1 | *Sz. pombe* | h$^+$; *ade6*-M216; *leu1-32; ura4-D18; hsp90ΔKanMX*; pMR477 (*hsp90$^+$/ura4$^+$*) | This study |

*S. cerevisiae* strains are isogenic to BY4741
*Sz. pombe* strains are isogenic to ED668

**Table 2 | Plasmids used in this study**

| Plasmid | Host | Backbone | Insert | Ref |
|---|---|---|---|---|
| pMR325 | *S. cerevisiae* | p414-GPD | *HSP82* ORF | 33, 61 |
| pMR55W | *S. cerevisiae* | pRS314 | *HSC82* + 500 | 60 This study |
| pMR423 | *S. cerevisiae* | p414-GPD | hHsp90α ORF | 33 |
| pMR365 | *S. cerevisiae* | p414-GPD | hHsp90β ORF | 59 |
| pMR453 | *S. cerevisiae* | p414-GPD | CnHsp90 ORF | This study |
| pMR454 | *S. cerevisiae* | p414-GPD | EhHsp90 ORF | This study |
| pMR455 | *S. cerevisiae* | p414-GPD | PfHsp90 ORF | This study |
| pKG2 | *S. cerevisiae* | p414-GPD | SpHsp90 ORF | This study |
| pMR169 | *S. cerevisiae* | pRS313 | P$_{GAL10}$::luciferase | 62 |
| pMR477 | *Sz. pombe* | pUR18 | *hsp90$^+$* (−669/+316) | 63 This study |
| pMR496 | *Sz. pombe* | pSP1 | P$_{tdh1}$::MCS::T$_{CYC1}$ | 64 This study |
| pMR497 | *Sz. pombe* | pMR496 | *hsp90$^+$* ORF | This study |
| pMR498 | *Sz. pombe* | pMR496 | *HSP82* ORF | This study |
| pMR499 | *Sz. pombe* | pMR496 | *HSC82* ORF | This study |
| pMR500 | *Sz. pombe* | pMR496 | hHsp90α ORF | This study |
| pMR501 | *Sz. pombe* | pMR496 | hHsp90β ORF | This study |
| pSK59 | *E. coli* | pET28b | *HSP82* ORF | 59 |
| pSK92 | *E. coli* | pET28b | *AHA1* ORF | 59 |

Hsp90 has low affinity for ATP and extremely weak intrinsic ATPase activity. Indeed, for some time after its discovery, there were conflicting reports regarding what role, if any, ATP had in Hsp90 function[23] (reviewed in:[19]). In 1997 crystallography showed that Hsp90 did indeed bind ATP[24]. The authors proposed that Hsp90 activity depended on ATP hydrolysis as DNA gyrase B, due to structural similarities, but could not rule out a reaction mechanism based on nucleotide exchange[24]. Two papers published in 1998 concluded ATP hydrolysis was crucial[25,26]. Using the structural data, both groups engineered mutations in *Saccharomyces cerevisiae* Hsp82 that either blocked ATP hydrolysis (E33A) or binding (D79N) and reported that either mutant was lethal when expressed as the only Hsp90 in *S. cerevisiae*. They concluded that both ATP binding and hydrolysis were required for in vivo Hsp90 function. The following year it was reported that hydrolysis-defective Hsp90 was unable to fully mature progesterone receptor in a reconstituted in vitro system, in agreement with the in vivo studies[27]. However, the exchange mechanism hypothesis was not directly tested. In the decades since, the requirement of ATP hydrolysis for Hsp90 to function has become almost universally accepted and is the cornerstone of current models of Hsp90 activity[19–22].

In 2016, however, it was reported that *S. cerevisiae* expressing the E33A variant of Hsp82 were viable, but defective in several Hsp90-specific functions such as activation of heterologously expressed mammalian v-SRC kinase and glucocorticoid receptor (GR)[13]. The authors characterized their finding that Hsp82-E33A functioned in vivo as unexplainable, since it was incompatible with the model of Hsp90 function in which ATP hydrolysis is required for Hsp90 to release the mature client. The authors also found that ATP hydrolysis alone was not sufficient for the essential function of Hsp90 in *S. cerevisiae*, as

mutations that blocked conformational rearrangements yet hydrolyze ATP were lethal[13]. Thus, Hsp90's conformational rearrangements themselves, independently of ATP hydrolysis, were needed for its essential function in vivo. These findings are in general agreement with the nucleotide exchange mechanism suggested earlier[24]. Nonetheless, subsequently published models of the Hsp90 reaction cycle state that its completion depends on ATP hydrolysis[20–22]. More recent findings on the regulation of Hsp90 activity by the co-chaperone Aha1, however, have provided additional evidence in support of the nucleotide exchange mechanism[28,29]. Furthermore, cytosolic and organelle-resident human Hsp90 paralogs possess hydrolysis-independent activities[30–32]. Thus, the exact role of ATP hydrolysis in Hsp90 function remains uncertain.

Recently we reported on forward mutations in human Hsp90α that improved its ability to support viability of *S. cerevisiae*[33]. We subsequently tested whether the forward mutations bypassed hHsp90α's requirement for ATP hydrolysis by combining them with the hydrolysis defective mutation. Unexpectedly, we observed that control strains expressing the hydrolysis defective versions grew better than those expressing wild type hHsp90α. We thus expanded our study of the requirement for ATP hydrolysis and nucleotide binding by using different Hsp90 orthologs and biological systems.

Here, we confirm that *S. cerevisiae* cells expressing the ATP hydrolysis-defective E33A variant (EA) of Hsp82 are viable, but cells expressing the nucleotide binding-defective D79N variant (DN) are not. Effects of these mutations are conserved in various eukaryotic Hsp90 orthologs when expressed in either *S. cerevisiae* or *Schizosaccharomyces pombe*, a system we developed and describe here. We identify second-site suppressors of EA-mediated defects that restore normal function in vivo, the effects of which are conserved across Hsp90 orthologs and host species. Importantly, the suppressor mutations do not restore ATP hydrolysis. Our findings show that ATP hydrolysis is dispensable, but nucleotide binding is required, for Hsp90 function in two evolutionarily distant biological systems. Our findings strongly support a mechanism of Hsp90 activity in which nucleotide exchange occurs frequently enough to reach the ADP-bound state without ATP hydrolysis. Switching between nucleotide states by either exchange or hydrolysis promotes the conformational rearrangements needed for Hsp90's essential and non-essential functions in vivo.

## Results

### ATP hydrolysis, but not nucleotide binding, is dispensable for Hsp90 function in vivo

*Saccharomyces cerevisiae* is widely used to study Hsp90 function in vivo. In our system, the two Hsp90 genes, *HSC82* and *HSP82*, have been deleted from the genome and essential Hsp90 function is supplied by an *HSP82* allele on an *URA3*-marked plasmid. This strain, MR1075 (see Table 1), was first transformed with *TRP1*-marked plasmids that encode test Hsp90 constructs driven by a strong constitutive promoter (see Table 2). The test Hsp90 can be either a mutant yeast Hsp90 or an orthologous Hsp90 from any species. Patches of pooled transformants are then grown on medium containing uracil but lacking tryptophan: if the test Hsp90 on the *TRP1* plasmid can complement

**Table 3 | Amino acid positions of mutants by Hsp90 ortholog**

| Hsp90 | Binding defective (DN) | Hydrolysis defective (EA) | EA suppressor 1 (TA) | EA suppressor 2 (EK) |
|---|---|---|---|---|
| Hsp82 | D79N | E33A | T171A | E372K |
| Hsc82 | D79N | E33A | T171A | E368K |
| hHsp90α | D93N | E47A | T184A | E392K |
| hHsp90β | D88N | E42A | T179A | E384K |
| SpHsp90 | D80N | E34A | T172A | E367K |
| CnHsp90 | D79N | E33A | | |
| EhHsp90 | D90N | E44A | | |
| PfHsp90 | D79N | E33A | | |

essential Hsp90 function, then under these conditions the parental *URA3* plasmid can be lost. Loss events are selected for by replica-plating to medium containing 5′-fluoro-orotic acid (FOA), which kills cells expressing Ura3. Thus, growth on FOA-containing medium reflects the ability of the test Hsp90 to complement essential Hsp90 function. Infrequent single colonies (papillae) that arise from rare events from a patch of otherwise dead cells on FOA plates are not evidence of complementation.

We transformed MR1075 with plasmids expressing Hsp90s from *S. cerevisiae* (Hsp82 and Hsc82), human (hHsp90α and hHsp90β), *Cryptococcus neoformans* (CnHsp90), *Entamoeba histolytica* (EhHsp90), *Plasmodium falciparum* (PfHsp90) or *Sz. pombe* (SpHsp90) (Table 2). In addition to the wild type Hsp90s, we also included mutant versions that were defective in ATP hydrolysis, called EA (analogous to E33A in Hsp82), and versions that were defective in nucleotide binding called DN (analogous to D79N in Hsp82). For clarity, mutations are referred to herein by the amino acid single letter codes since the position numbers of these conserved residues in the various Hsp90s are different (see Table 3).

All the wildtype Hsp90s complemented essential *S. cerevisiae* Hsp90 function, since cells expressing these Hsp90s grew on FOA (Fig. 1a, left column; Supplementary Fig. 1A). As previously observed, hHsp90α did not complement as well as hHsp90β, as seen by much weaker growth on FOA[33–35]. While CnHsp90 and EhHsp90 complemented *S. cerevisiae* Hsp90 function well, PfHsp90 complemented less well, similarly to hHsp90α. The EA versions of all the Hsp90s except SpHsp90 also complemented essential *S. cerevisiae* Hsp90 function, whereas the DN versions of all of them did not (Fig. 1a, middle and right columns; Supplementary Fig. 1A). The EA mutation had a stronger negative effect in hHsp90β and EhHsp90 than the other Hsp90 orthologs. In contrast to the other Hsp90s, the EA mutation improved growth of cells on FOA expressing hHsp90α and PfHsp90.

The conservation of EA and DN effects in different Hsp90 orthologs demonstrates that nucleotide binding but not ATP hydrolysis is required for essential Hsp90 function in *S. cerevisiae*. It was possible that *S. cerevisiae* itself was unique in not requiring Hsp90 ATP hydrolysis. To address this possibility, and to understand why SpHsp90-EA failed to complement *S. cerevisiae*, we developed a *Sz. pombe* system that also uses growth on FOA as a readout for essential Hsp90 function (see "Methods" and Table 1). Although both organisms are yeasts they are quite distant evolutionarily, diverging some 330-420 million years ago. In fact, each yeast is of similar evolutionary divergence to humans as to each other[36, 37]. Using this system, we observed that SpHsp90-EA supported viability of *Sz. pombe* cells (Fig. 1b). While wild type *S. cerevisiae* Hsp82 and Hsc82 complemented *Sz. pombe* Hsp90 function, the EA and DN versions did not (Fig. 1b), which mirrored the results of SpHsp90's in *S. cerevisiae*. Thus, ATP hydrolysis is not required for essential Hsp90 function in either *S. cerevisiae* or the evolutionarily distant *Sz. pombe*.

We quantified the effect of EA in Hsp82 and Hsc82 on growth of *S. cerevisiae* under optimal conditions in liquid culture (Fig. 1c). Cells expressing either Hsp82-EA or Hsc82-EA grew slower than their wild type counterparts. The EA mutation had a stronger detrimental effect in Hsc82, even though steady-state abundance of the mutant proteins were similar (Supplementary Fig. 1B). Using the previously reported relationship between growth rate and Hsp82 expression level[38], and correcting for the increased expression level of the EA mutants (Supplementary Fig. 1B, "Methods"), we estimated that Hsp82-EA and Hsc82-EA were 1.8% and 0.8% as efficient as wild type, respectively. Also, cells expressing either Hsp82-EA or Hsc82-EA were hypersensitive to the Hsp90-specific ATP-competitive inhibitor radicicol (Fig. 1d; Supplementary Fig. 1C).

## The EA mutation alters Hsp82 conformation changes in response to different nucleotides

To understand how Hsp90-EA supported growth, we used fluorescence-based techniques to investigate whether EA altered nucleotide-mediated conformational rearrangements in Hsp82, which are essential to its function in vivo. Rearrangements associated with formation of the closed-clamp such as N-M domain docking, lid repositioning and β-strand swapping are observed via photoinduced electron transfer (PET)[7,28,33]. A cysteine and a tryptophan residue are introduced at defined positions depending on which movement is to be measured. The cysteine is then labeled with a fluorophore. Addition of non-hydrolysable ATP analogs such as AMP-PNP or ATP-γ-S induce rearrangements that bring the fluorophore closer to the engineered tryptophan, quenching the fluorescence (Fig. 2a; Supplementary Fig. 2A, D).

The use of non-hydrolysable analogs in the PET assays is common, findings reproduced here (Fig. 2b, solid and dashed lines). However, we are aware of only one study that used ATP in a single-molecule PET experiment[8]. We found that addition of ATP to wild type Hsp82 resulted in little to no N-M docking (Fig. 2b, dotted line). Similar results were obtained in lid repositioning and β-strand swapping PET experiments (Supplementary Fig. 2). Thus, non-hydrolysable analogs AMP-PNP and ATP-γ-S induced several conformational rearrangements in wild type Hsp82 but ATP did not. These results agree with reports that showed different effects of ATP versus non-hydrolysable analogs on changes in Hsp90's conformation[6, 13–16].

In contrast to wild type Hsp82, N-M docking occurred at a normal rate when ATP was added to Hsp82-EA (Fig. 2c, dotted line). However, no docking was observed when AMP-PNP or ATP-γ-S were added (Fig. 2c, solid and dashed lines). Again, we observed similar results in lid repositioning and β-strand swapping PET (Supplementary Fig. 2C, F).

Hsp82 dimer closing also can be observed using fluorescence resonance energy transfer (FRET) with labels in the M and N domains of opposing protomers (Fig. 2d)[13,39,40]. As reported, we observed a FRET signal upon addition of AMP-PNP to labeled Hsp82 proteins (Fig. 2e, solid line). However, we observed very little FRET signal in wild type Hsp82 upon addition of ATP (Fig. 2e, dotted line), in line with previous reports[6,13,15]. Similar to what we observed in the PET experiments, we observed significantly more FRET signal in Hsp82-EA upon addition of ATP than AMP-PNP (Fig. 2f). Taken together, ATP and the analogs had the opposite effects on the conformational rearrangements of Hsp82-EA compared to wild type.

## *S. cerevisiae* and *Sz. pombe* cells expressing ATP hydrolysis-deficient Hsp90s are defective in non-essential functions

Although *S. cerevisiae* expressing Hsp82-EA or Hsc82-EA were viable, they were defective in activation of two non-native clients, mammalian v-SRC and GR[13]. ATP hydrolysis might therefore be required for the maturation of certain clients. We tested this idea by using two assays based on pathways that are known to be regulated by, and therefore are natural clients of, *S. cerevisiae* Hsp90: maintenance of cell wall integrity (CWI) and galactose-regulated gene induction (GAL).

The CWI pathway is induced in *S. cerevisiae* by cell wall stress, such as high temperatures or exposure to cell wall-damaging agents such as

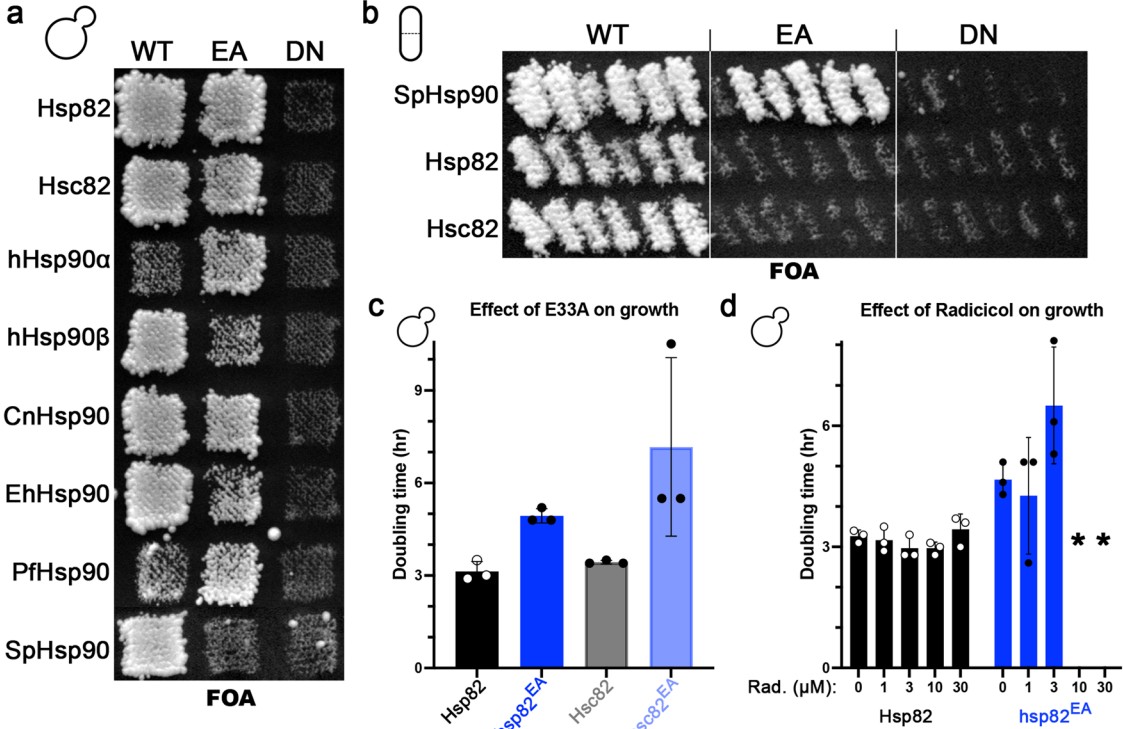

**Fig. 1 | ATP hydrolysis, but not binding, is dispensable for viability of *S. cerevisiae* or *Sz. pombe*. a** Plasmid shuffle in *S. cerevisiae* (budded cell). Hsp90s from *S. cerevisiae* (Hsp82, Hsc82), human (hHsp90α and hHsp90β), *C. neoformans* (CnHsp90), *E. histolytica* (EhHsp90), *P. falciparum* (PfHsp90) and *Sz. pombe* (SpHsp90) complemented the essential function of *S. cerevisiae* Hsp90 (left column, wt), as denoted by uniform growth of patches on FOA plates. The hydrolysis defective (EA) versions of all the Hsp90s except SpHsp90 also complemented *S. cerevisiae* Hsp90 essential function (middle column). None of the nucleotide binding defective (DN) Hsp90s could support *S. cerevisiae* viability (right column). **b** Plasmid shuffle in *Sz. pombe* (fission cell). The hydrolysis defective (EA) version of

*Sz. pombe* Hsp90 (SpHsp90) supported viability of *Sz. pombe* cells, but the nucleotide binding defective (DN) version did not (top row). Wild type *S. cerevisiae* Hsp90s, but not the EA or DN versions, supported viability of *Sz. pombe* (second and third rows). **c** *S. cerevisiae* cells expressing EA versions of Hsp82 or Hsc82 grew slower than wild type. **d** *S. cerevisiae* cells expressing Hsp82-EA were hypersensitive to the Hsp90 inhibitor radicicol (concentration range 0 to 30 μM). Asterisks denote conditions under which no growth was observed. In (**c**) and (**d**), bars are the average doubling times of three biological replicates (circles) and the error bars are the standard deviation. Source data are provided as a Source Data file.

calcofluor white (CFW). Stress signals at the cell wall or plasma membrane are transmitted via a MAP kinase cascade that ultimately phosphorylates the transcription factor Rlm1, which then activates genes involved in cell wall repair and maintenance (reviewed in:[41]). In *S. cerevisiae* the final kinase in the cascade, Slt2, is a known Hsp90 client[42–44], and in *Aspergillus fumigatus* several other conserved CWI components were shown to be Hsp90 clients[45]. One of these was PkcA, the ortholog of *S. cerevisiae* Pkc1, the initial kinase in the cascade.

Defects in CWI signaling result in osmolyte-remediated temperature sensitivity, leaky cell walls and sensitivity to CFW[41]. *S. cerevisiae* expressing Hsp82-EA or Hsc82-EA exhibited all of these CWI phenotypes (Fig. 3a). *Sz. pombe* cells expressing SpHsp90-EA had an osmolyte-remediated temperature sensitivity phenotype, which strongly suggested conservation of EA-specific phenotypes (Fig. 3b). We observed a decrease in Slt2 phosphorylation and corresponding reduced induction of Slt2 protein, which is itself a downstream transcriptional target of phosphorylated Slt2, in *S. cerevisiae* expressing Hsp82-EA or Hsc82-EA (compared to wild type) upon cell wall stress. These results indicated that *S. cerevisiae* Hsp90 acted upstream of Slt2 as it does in *Aspergillus*[45] (Fig. 3c). To test this idea, we observed the localization of GFP-tagged Pkc1. Proper localization of Pkc1 to sites of polarized growth and bud necks has been linked to its proper function[46]. Indeed, we found that Pkc1-GFP was mostly cytosolic and rarely observed at growing tips or bud necks in *S. cerevisiae* cells expressing Hsp82-EA (Fig. 3d). Together with the recent findings in *Aspergillus*[45], our results show that Hsp90's role in CWI is conserved across fungal species. Thus, hydrolysis defective Hsp90s were unable

to properly chaperone native CWI clients, which manifested as CWI phenotypes.

The *S. cerevisiae GAL* genes are required for the utilization of galactose. *GAL* genes are not expressed in the absence of galactose and are repressed in the presence of glucose. Hsp90 regulates the *GAL* system at both the transcriptional and protein levels[47–49]. *S. cerevisiae* expressing wild type or EA versions of Hsp82 or Hsc82 and harboring a galactose-inducible luciferase reporter (Table 2) were grown in non-repressible liquid medium (see "Methods"). After measuring basal luciferase activity, galactose was added to the cultures and luciferase activity was measured again after 2 h. Some cultures expressing wild type Hsp82 or Hsc82 were pretreated with a sublethal concentration (60 μM) of radicicol for 1 h before induction. No induction of luciferase was detected in radicicol-treated wild type cells or cells expressing Hsp82-EA or Hsc82-EA (Fig. 3e). Together with the cell growth and CWI results, these findings agree with the idea that Hsp90 ATP hydrolysis is required for some, but not all clients. However, they also support the notion that a certain level of Hsp90 activity that cannot be supplied by Hsp90-EA is required for non-essential functions.

## Second-site suppressor mutations rescue slow growth of cells expressing EA

The inability of Hsp90-EA to function properly in CWI or *GAL* could be explained by either a client specific requirement for ATP hydrolysis or because the EA mutation reduces general Hsp90 activity below a threshold that is needed for non-essential processes such as CWI and *GAL*. To address this question, we screened for second-site suppressors

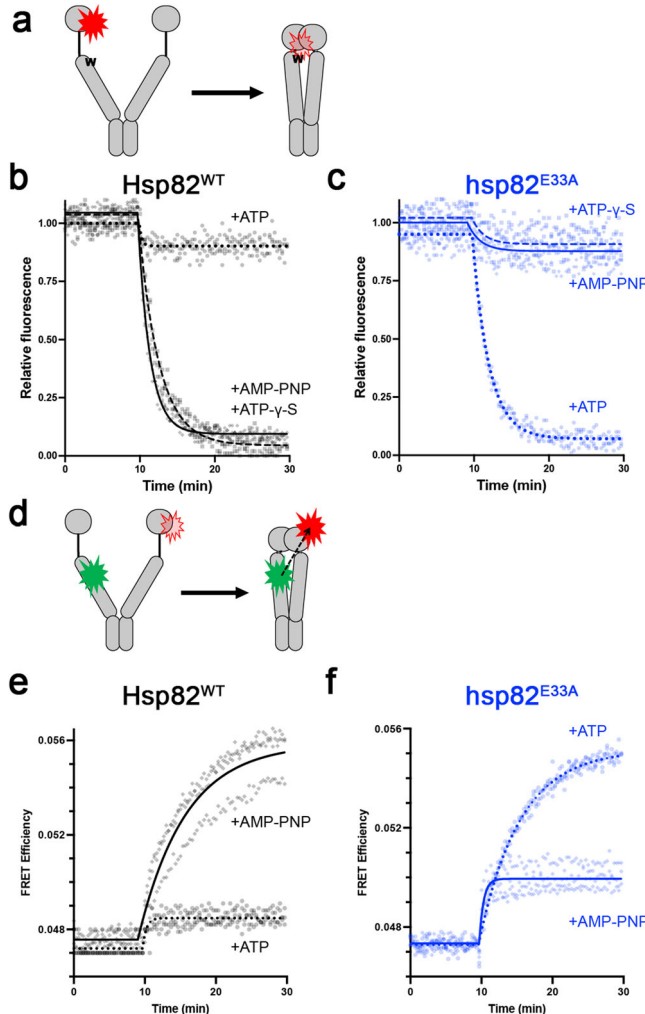

**Fig. 2 | ATP binding alone does not induce the closed camp in wild type Hsp82 and the E33A mutation alters conformational responses to nucleotides. a** In PET experiments, formation of the closed-clamp brings a fluorophore in the N domain close to an engineered tryptophan (W) in the M domain which quenches the fluorescence. **b** Wild type Hsp82 proteins formed the closed-clamp upon addition (at T = 10) of AMP-PNP (black diamonds are data points and black solid line is the fit) or ATP-γ-S (black squares and dashed line), but not ATP (black circles and dotted line). **c** Conversely, Hsp82-EA proteins formed the closed-clamp in the presence of ATP (blue circles and dotted line) but not AMP-PNP (blue diamonds and solid line) or ATP-γ-S (blue squares and dashed line). **d** In FRET, a green fluorophore in the M domain excites a red fluorophore in the N domain of the opposing protomer upon formation of the closed-clamp. **e** Formation of the closed-clamp in wild type occurred upon addition of AMP-PNP (black diamonds and solid line) but not ATP (black circles and dotted line). **f** Formation of the closed-clamp in Hsp82-EA occurred upon addition of ATP (blue circles and dotted line) but not AMP-PNP (blue diamonds and solid line). For all PET and FRET, lines are the exponential decays fits of the average of three independent experiments. Source data are provided as a Source Data file.

of EA defects. We reasoned that suppressor mutations that improved Hsp90-EA's ability to provide essential Hsp90 function (as determined by improvement of growth rates over Hsp90-EA under optimal conditions) would not rescue Hsp90-EA defects in non-essential pathways if the ATP hydrolysis requirement was client specific. Based on the critical role E33 (or equivalent) plays in the hydrolysis reaction, we predicted that any suppressors would not restore ATP hydrolysis.

We chose hHsp90β-EA because the EA-mediated growth defect under optimal conditions was the most severe in hHsp90β (Fig. 1a; Supplementary Fig. 1A). We created a library of random mutations in

the plasmid encoding hHsp90β-EA and used it to transform MR1075 (see "Methods"). We then isolated faster growing colonies on FOA. After a secondary screen to eliminate false positives (see "Methods"), we identified several mutations that improved the ability of hHsp90β-EA to support *S. cerevisiae* growth. We chose mutations in two conserved residues, T179A and E384K, for initial study (Fig. 4a). In a separate screen to identify suppressors of hHsp90α-EA defects, we twice isolated E392K, which is analogous to hHsp90β E384K. Finding the same suppressor in two different Hsp90 orthologs demonstrated conservation of function. As above, the amino acid position numbers of the suppressors are omitted and are referred to as TA and EK (see Table 3).

Residue T179 is inside the hHsp90β nucleotide binding pocket and is situated very close to D88 (analogous to Hsp82 D79, see Table 3), which forms a hydrogen-bond with the adenosine ring of bound nucleotides (Fig. 4a). This residue in the context of a different Hsp90 ortholog was shown to contact geldanamycin and ATP[40, 50]. It is conceivable that an alanine at position 179 disrupts the stability of the bound nucleotide by removing the polar hydroxyl moiety near the adenosine ring. E384 is in the M domain but very close to the lid structure that repositions over the nucleotide in the closed-clamp conformation (Fig. 4a)[10,51]. A charge reversal in this position might alter lid stability. TA and EK each improved the ability of hHsp90β-EA to support *S. cerevisiae* essential Hsp90 function, but EK had a noticeably stronger suppressive effect (Fig. 4b).

To determine whether the suppressive effect of the forward mutations on EA was conserved in other Hsp90s, we combined TA or EK with EA in Hsp82, Hsc82, SpHsp90 and hHsp90α for expression in *S. cerevisiae* (Fig. 4c; Supplementary Fig. 3A). We quantified the effect of the suppressors on the EA-mediated growth defect in liquid culture under optimal conditions (Fig. 4d). As observed in hHsp90β, EK suppressed the Hsp82-EA growth defect better than TA. The expression levels of the rescued mutants were comparable to wild type (Supplementary Fig. 3B). The TA suppressor improved the relative efficiency of Hsp82-EA by approximately fivefold (Hsp82-EATA relative efficiency was 10%), while cells expressing Hsp82-EAEK were as efficient as wild type with respect to growth (Fig. 4c, d). While EK was independently identified as a suppressor of hHsp90α-EA (see above), cells expressing hHsp90α-EATA grew more slowly on FOA than those expressing hHsp90α-EA (Fig. 4c; Supplementary Fig. 3A). In contrast to the other Hsp90 orthologs tested, TA or EK barely rescued the inability of *Sz. pombe* Hsp90-EA to support growth of *S. cerevisiae* (Fig. 4c; Supplementary Fig. 3A).

TA or EK improved the ability of *S. cerevisiae* or human Hsp90-EA, but not *Sz. pombe* Hsp90-EA, to support *S. cerevisiae* growth. To help understand this discrepancy, we expressed the same Hsp90s in *Sz. pombe*. SpHsp90-EATA and -EAEK supported growth of *Sz. pombe* well (Fig. 4e), in contrast to their inability to support *S. cerevisiae* viability (Fig. 4c). The EA or EAEK versions of Hsp82 or Hsc82 failed to support *Sz. pombe* viability, while *Sz. pombe* cells expressing Hsp82-EATA or Hsc82-EATA grew but very poorly. As above, these results mirror *S. cerevisiae* cells expressing SpHsp90 variants. However, all versions of the human Hsp90s behaved similarly in both systems (Fig. 4c, e). Thus, there appears to be some component missing that allows hydrolysis-defective versions of *S. cerevisiae* Hsp90s to complement *Sz. pombe*, and vice versa. This barrier does not exist between the Hsp90s of human and *S. cerevisiae* or human and *Sz. pombe*, which likely reflects the evolutionary divergence among the three species. Taken together, the effects of second-site suppressors identified in hHsp90β are mostly conserved across Hsp90 orthologs and host systems.

### The suppressor mutations rescue the CWI and *GAL* defects of Hsp82-EA in *S. cerevisiae*, and SpHsp90-EA temperature sensitivity in *Sz. pombe*

Both TA and EK ameliorated Hsp82-EA radicicol hypersensitivity, which agreed with the idea that the suppressors provided a general

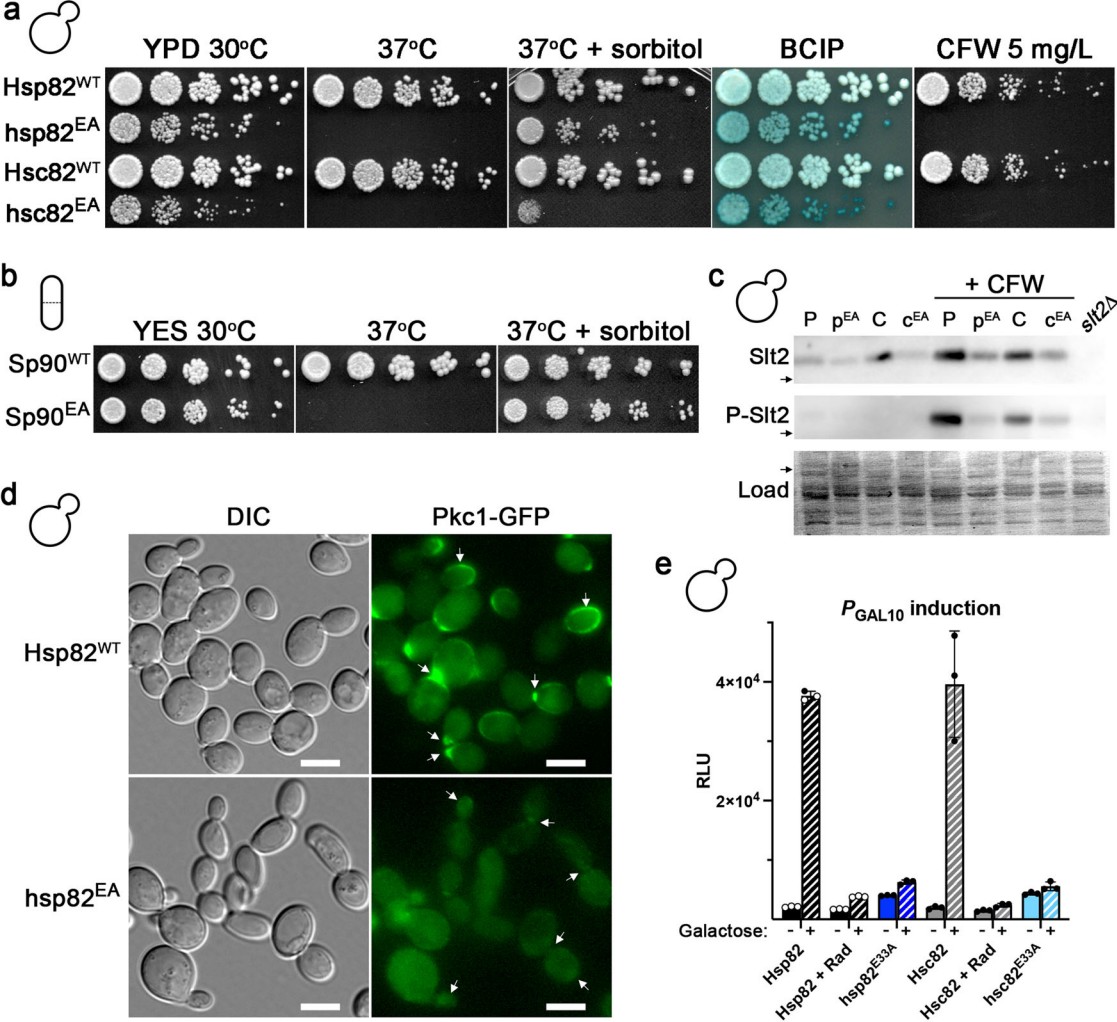

**Fig. 3 | Cells expressing hydrolysis defective Hsp90 display conditional phenotypes. a** *S. cerevisiae* cells expressing the EA versions of Hsp82 or Hsc82 were osmo-remediated temperature sensitive (compare growth at 37 °C with and without sorbitol), had cell walls that leaked phosphatase at optimal temperatures (as denoted by bluer color of EA cells after BCIP overlay, see "Methods") and were sensitive to cell wall damaging agents such as calcofluor white (CFW). **b** *Sz. pombe* expressing SpHsp90-EA was osmo-remediated temperature sensitive.
**c** Representative western blots (of three independent experiments) showing levels of Slt2 protein (top) or phosphorylated Slt2 (P-Slt2, middle) in log-phase *S. cerevisiae* cells expressing wild type Hsp82 (P) or Hsc82 (C) or the E33A versions (p^EA, c^EA),

without (left 4 lanes) and with 60 min of CFW treatment (right 4 lanes). Arrows denote the location of the 50 kD molecular weight marker. **d** GFP-tagged Pkc1, the initial kinase in the CWI cascade, is localized to sites of polarized growth and bud necks (arrows) in cells expressing wild type Hsp82 (top panels), but not Hsp82-E33A (bottom panels). Shown are cells that represent the results from three independent experiments. Scale bars are 5 μm. **e** *S. cerevisiae* cells treated with radicicol or expressing Hsp82-EA or Hsc82-EA did not express a luciferase reporter of *GAL* activation upon addition of galactose. Bars are the average of three biological replicates (circles) and the error bars are the standard deviation. Source data are provided as a Source Data file.

enhancement of Hsp90-EA function (Fig. 5a). To address whether non-essential Hsp90 functions such as CWI and *GAL* had an ATP hydrolysis requirement, we asked whether the suppressors rescued Hsp90-EA defects in these pathways. Both TA and EK suppressed the EA-mediated CWI defect, but the suppressive effect of EK was noticeably stronger than TA (Fig. 5b; Supplementary Fig. 4). Likewise, EK improved the ability of EA to induce the galactose-specific luciferase reporter, although not to wild type levels (Fig. 5c). Consistent with its reduced ability to rescue EA defects, the TA mutation failed to rescue the EA defect in *GAL* induction.

The suppressive effect of TA and EK on EA defects in non-essential functions were not specific to *S. cerevisiae*, as these mutations also rescued SpHsp90-EA mediated temperature sensitivity in *Sz. pombe* (Fig. 5d). Taken together, these results demonstrate that the failure of Hsp90-EA to activate CWI or *GAL* was due to an overall reduction in function that was restored to varying degrees by the TA or EK suppressors.

## The suppressors of EA do not restore ATP hydrolysis but increase nucleotide exchange

We assumed that the TA and EK suppressors did not restore ATP hydrolysis. To verify this assumption was correct, we purified the mutant Hsp82 proteins and tested whether TA or EK restored ATP hydrolysis when combined with EA. As expected, we did not detect any ATP hydrolysis in reactions containing Hsp82-EA, Hsp82-EATA or Hsp82-EAEK (Fig. 6a). Aha1, a stimulator of Hsp90 ATPase, was added in separate reactions to maximize any ATP hydrolysis. However, none was detected in these reactions either (Fig. 6a). Thus, the suppressive effect of TA and EK on EA in vivo was not due to restoration of ATP hydrolysis.

If hydrolysis-deficient Hsp90 functioned in vivo through a nucleotide exchange mechanism, and the suppressors facilitated exchange, then a mutation that reduces binding to ADP specifically would be expected to exacerbate the hydrolysis-defective phenotype. To test this hypothesis we introduced K98A, a mutation that has been

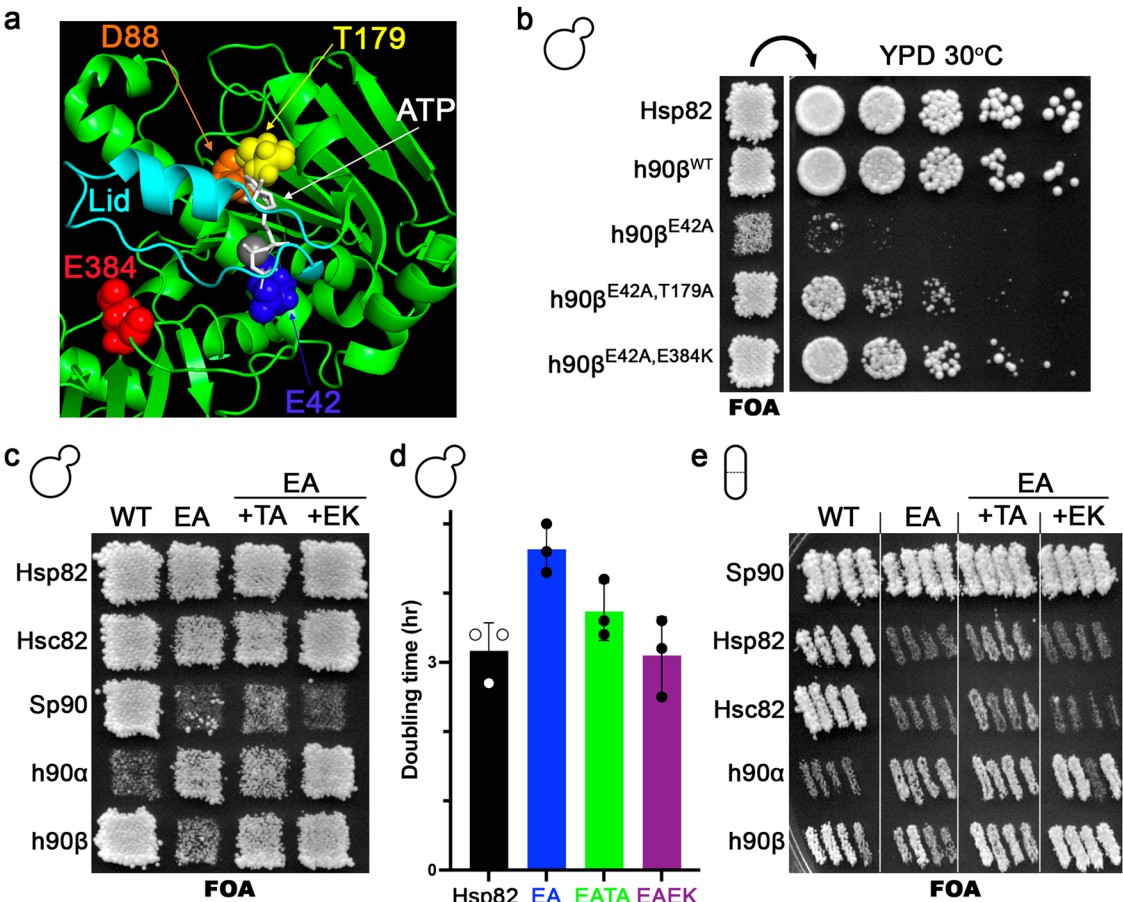

**Fig. 4 | Second site mutations rescue the hHsp90β-EA growth defect and their effects are conserved. a** Suppressors of the hHsp90β-EA growth defect are in or near the ATP binding pocket. T179 (yellow) is next to D88 (orange) and interacts with the adenosine ring of bound nucleotides (white). E384 (red) is in the M domain but comes very close to the lid structure (cyan) in closed Hsp90. E42 (blue) and Mg$^{++}$ (gray sphere) are also shown for reference. hHsp90β structure is from PDB ID 5fwk[51]. **b** T179A and E384K improved the ability of hHsp90β-EA to support viability of *S. cerevisiae*. **c** EK improved the growth of *S. cerevisiae* cells expressing hHsp90α-

EA but not SpHsp90-EA. TA modestly rescued SpHsp90-EA but not hHsp90α-EA. **d** TA and EK rescued the EA slow growth defect of *S. cerevisiae* cells expressing EA. Cells expressing EAEK grew as fast as cells expressing wild type Hsp82. Bars are the average doubling times of three biological replicates (circles) and the error bars are the standard deviation. Source data are provided as a Source Data file. **e** In *Sz. pombe*, the effect of TA and EK on improving growth of human Hsp90s was similar as in *S. cerevisiae*, while the suppressors did not improve growth of *Sz. pombe* cells expressing EA versions of *S. cerevisiae* Hsp90s.

shown to decrease affinity specifically for ADP[6], into Hsp82 and Hsp82-EA. As expected, *S. cerevisiae* cells expressing the Hsp82-E33A,K98A double mutant barely grew under optimal conditions while cells expressing wild type Hsp82, Hsp82-EA or Hsp82-K98A grew normally (Fig. 6b). Thus, binding to ADP was important for Hsp82-EA function in vivo. Since Hsp82-EA could not attain the ADP state through hydrolysis it must have done so through exchange.

## Discussion

Here, we confirm that *S. cerevisiae* cells expressing ATP hydrolysis defective Hsp82 are viable. These findings agree with a recent report[13] but conflict with earlier papers[25,26,52]. We cannot explain the discrepancy, except to note that two earlier studies fused hexa-histidine tags onto their Hsp90s, which may have allele-specific effects[25,26]. A more recent growth competition study among all possible mutations in the Hsp82 N-domain found every amino acid except glutamate at position 33 was unfit[52]. However, the mutants were expressed at a low level which reduces Hsp90 activity[38] and grown in a competitive environment, in contrast to our system where each mutant is tested for its individual ability to support growth. Importantly, our work and Mishra et al. together show that while ATP hydrolysis is not required, it is selected for, meaning that ATP hydrolysis provides an evolutionary advantage.

We further showed that Hsp90s from human and other species containing the hydrolysis defective mutation complemented growth of *S. cerevisiae*. Thus, the lack of a hydrolysis requirement is not unique to Hsp82. The dispensability of ATP hydrolysis was also not unique to *S. cerevisiae*, as hydrolysis-defective *Sz. pombe* and human Hsp90s supported viability of *Sz. pombe*. Since *S. cerevisiae*, *Sz. pombe* and humans are all of approximately equal evolutionary divergence from one another[36,37], our results show that this property of the Hsp90 protein is conserved across hundreds of millions of years of evolution. While our work using fungal species does not address whether ATP hydrolysis is required for essential Hsp90 functions in metazoans, any such requirement might be expected to be cell type specific since hydrolysis defective human Hsp90s function in both *S. cerevisiae* and *Sz. pombe*, eukaryotes that do not require hydrolysis.

We found that ATP did not induce changes in wild type Hsp82 conformation the way non-hydrolysable analogs did, which agrees with observations by other groups using various techniques[6,13–16]. Interestingly, we found that EA flipped the effects of ATP and the analogs on induction of conformational rearrangements. Since the analogs trap Hsp82 in the pre-hydrolysis state, we expected that ATP would have a similar effect in EA as the analogs in wild type, since in both cases no hydrolysis can occur. But the finding that the analogs failed to induce conformational changes in EA, which were also

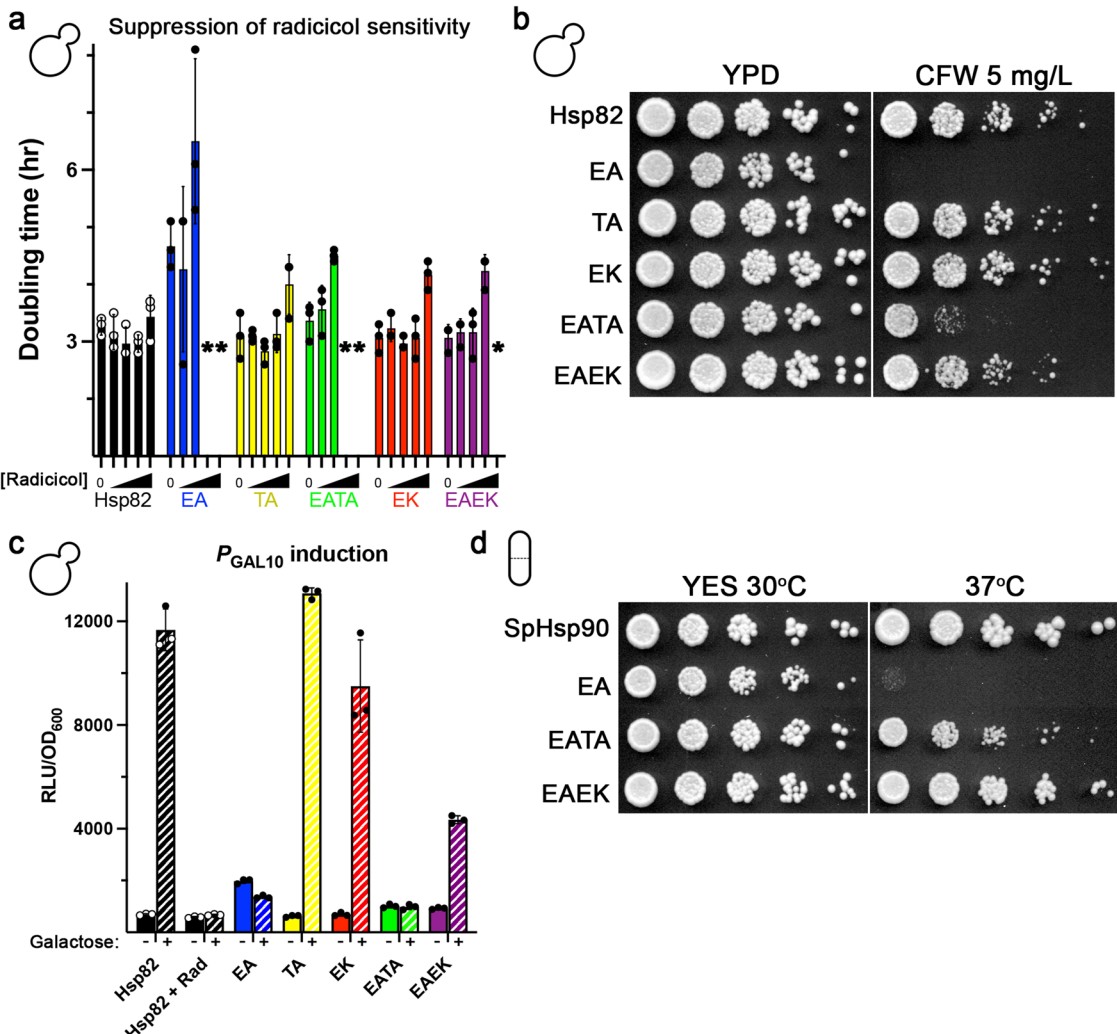

**Fig. 5 | The TA and EK mutations rescue EA-mediated defects in conditional phenotypes to varying degrees. a** *S. cerevisiae* cells expressing EATA or EAEK versions of Hsp82 were less sensitive to radicicol (concentrations: 0, 1, 3, 10 and 30 μM) than cells expressing Hsp82-EA. Asterisks (*) denote conditions under which no growth was observed. **b** *S. cerevisiae* cells expressing Hsp82-EATA or -EAEK were less sensitive to CFW than cells expressing Hsp82-EA. **c** EK partially rescued the Hsp82-EA *GAL* induction defect in *S. cerevisiae*. **d** *Sz. pombe* cells expressing EATA or EAEK versions of *Sz. pombe* Hsp90 were not temperature sensitive. In (**a**) and (**c**), bars are the average of three biological replicates (circles) and the error bars are the standard deviation. Source data are provided as a Source Data file.

conditions where no hydrolysis can occur, was unexpected. While we cannot definitively rule out the unlikely possibility that E33A selectively blocks binding of non-hydrolysable analogs, at this time the mechanism underlying our observations remains unclear. Regardless, it is obvious that non-hydrolysable analogs and ATP have different effects on the movements of wild type Hsp90 and that EA alters the conformational response to the different nucleotides.

One explanation might be related to the position of the nucleotide phosphates within the binding pocket. It was shown that the gamma phosphate of bound ATP is highly mobile[24]. E33 is near the gamma phosphate and plays an important role in the hydrolysis reaction by coordinating the attacking water molecule and $Mg^{++}$ ion (see Fig. 4a)[25,26,51]. Alanine in place of glutamate at residue 33 replaced negative charge with increased hydrophobicity, which might have influenced the gamma phosphate to facilitate adoption of the position needed to trigger conformational rearrangements. This explanation may also underpin the flipped response to ATP and the analogs by EA. If the positioning of the ATP gamma phosphate is important for closing, then the analogs, which exhibit different electrostatic profiles compared to ATP, might be mimicking a conformation of ATP that triggers closing in wild type Hsp90 but not Hsp90-EA.

Taken together, conformational rearrangements are not induced by the binding of ATP per se, but rather might be induced by specific positioning of the gamma phosphate of ATP within the binding pocket. This idea is in line with a recent report that proposed restriction of ATP's degrees of freedom within the binding pocket led to increased catalytic activity[40].

Since the suppressors of Hsp90-EA did not restore ATP hydrolysis, they likely functioned by increasing nucleotide exchange. Structural techniques such as NMR might shed light on exactly how the suppressors facilitated exchange. Because the dwell time of each conformational state is crucial for Hsp90 function in vivo[13], then non-suppressed Hsp90-EA must be able to exchange nucleotides at a rate that allows it to perform essential functions. However, when the cells were stressed or more demands of Hsp90 function were made, as in the case of radicicol treatment, CWI or *GAL*, the rate of exchange in Hsp90-EA was insufficient without the suppressor. This implies that return to the ADP state after binding ATP is important, and that Hsp90 does not need to hydrolyze ATP, so long as it is able to attain the ADP state through exchange at a rate that allows it to function. Indeed, cells expressing Hsp82-K98A, a mutant shown to be defective in binding ADP[6], had no phenotype because the ADP state

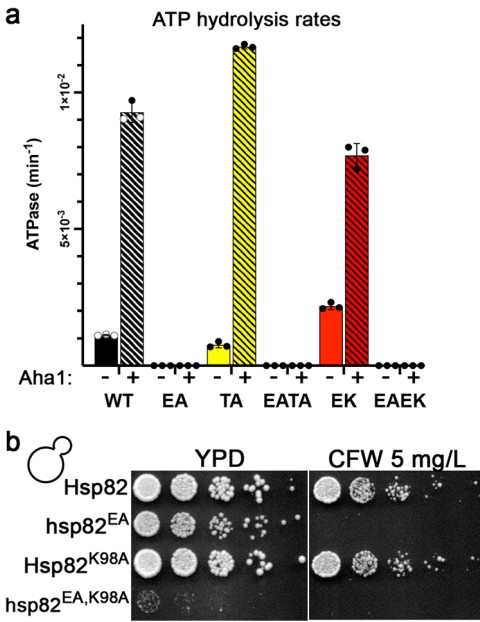

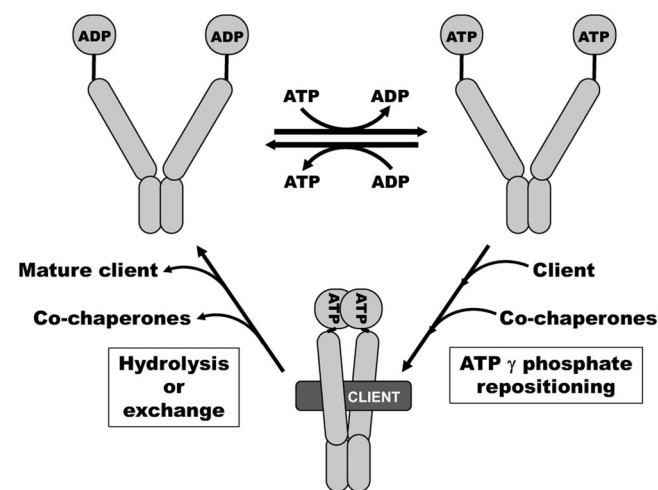

**Fig. 6 | The TA and EK mutations do not restore ATP hydrolysis and advancement to the ADP state is important for viability. a** The suppressor mutations TA or EK did not restore Hsp82 ATP hydrolysis when combined with EA. Bars are the average of three independent measurements (circles) and the error bars are the standard deviation. Source data are provided as a Source Data file. **b** *S. cerevisiae* cells expressing Hsp82-EA combined with the ADP-binding-specific mutation K98A grew very poorly at optimal temperature compared to wild type Hsp82, Hsp82-EA or Hsp82-K98A.

**Fig. 7 | An exchange model of Hsp90 activity.** Due to cellular nucleotide concentrations and its higher affinity for ADP, Hsp90 likely populates the open, ADP-bound conformation after hydrolysis (top left). Equilibrium between open ADP and ATP states, including ADP/ATP heterodimers (not depicted) is established due to the cellular ATP:ADP environment and regulation by co-chaperones. In the open ATP conformation, Hsp90 is primed for client binding (top right). Binding of the client, co-chaperones and, we propose, the proper positioning of the gamma phosphate of ATP drives formation of the closed clamp (bottom). Hydrolysis or exchange returns Hsp90 to the ADP state (top left), releasing the mature client and co-chaperones. The return to the ADP state from the closed-clamp is essential, since cells expressing a mutant Hsp90 that could not hydrolyze ATP or bind ADP effectively were barely viable.

was achieved through hydrolysis, while combining K98A with EA impeded both routes to the ADP state and was nearly lethal. Our work supports a model in which nucleotide exchange is sufficient to drive advancement to the ADP state that is needed for Hsp90 to function (Fig. 7).

Most current models state that ADP, phosphate, co-chaperones and the client are released after hydrolysis and Hsp90 adopts an apo state before binding ATP to reset the cycle[19–22]. However, the cell contains nucleotides at millimolar concentrations. Therefore, in vivo, Hsp90's nucleotide binding sites are likely occupied at any given moment. Likewise, due to the higher affinity for ADP compared to ATP[53], ADP can be expected to be bound to Hsp90 after hydrolysis, as proposed[14] (Fig. 7, top left). Due to the cellular ATP:ADP ratio (about 4:1)[54] and the action of co-chaperones, an equilibrium of ADP- and ATP-bound open conformations, including ADP/ATP heterodimers, is established. When bound to ATP, Hsp90 remains in an open state that is competent to bind clients (Fig. 7, top right)[14]. Formation of the closed-clamp is driven by binding of co-chaperones, the client[51] and, we propose, the repositioning of the gamma phosphate of ATP to a conformation mimicked by the analogs (Fig. 7, bottom). Our biophysical data together with findings of others[6,13–16] suggest this step is rate limiting. Hydrolysis of ATP or an exchange of ATP for ADP then returns Hsp90 to the open ADP state (Fig. 7, top left), and the mature client and co-chaperones are released. We propose that ADP in the binding pocket triggers opening of the closed clamp. The return to the ADP state from the closed-clamp is essential, since the Hsp82$^{E33A,K98A}$ mutant that could neither hydrolyze ATP nor effectively bind ADP barely supported growth.

Although the lid structure is positioned over the nucleotide in the closed clamp conformation, there is a channel through which ATP could dissociate (Supplementary Fig. 5), suggesting that ATP to ADP exchange can occur in the closed-clamp conformation. This mechanism might underlie the observation that addition of ADP can quickly re-open AMP-PNP-treated Hsp90[33]. We speculate that due to Hsp90's 2.5-fold higher affinity for ADP[53], the cytosolic nucleotide concentration and the ATP:ADP ratio of 4:1[54], conditions might exist on a knife's edge where exchange in either direction can happen often enough to allow Hsp90 to function. The high abundance of Hsp90 may facilitate favorable outcomes in this regard.

While ATP hydrolysis is not essential for Hsp90 function in vivo, it is clear that it enhances Hsp90 function since Hsc82-EA and Hsp82-EA were 1–2% as efficient as wild type in maintaining viability and incapable of performing the non-essential functions we tested, unless combined with a suppressor. However, because the suppressed variants behaved normally in vivo but were still unable to hydrolyze ATP, we conclude that energy from ATP hydrolysis is not required for even non-essential Hsp90 functions. Given that Hsp90 ATP hydrolysis is inhibited by ADP at concentrations that approximate cellular nucleotide concentrations[6], Hsp90 ATP hydrolysis in vivo may be relatively rare, and require input of co-chaperones and clients to activate when needed.

What, then, is the specific role of Hsp90's ATP hydrolysis activity? One hypothesis is that it serves only to drive Hsp90 back to the ADP state, which we show to be important for Hsp90 function in vivo. Another possibility is that hydrolysis occurs to reset Hsp90 in the case of a failed client maturation event or a stuck client. Rather than being strictly necessary, a key role for ATP hydrolysis is that it provides a regulatory element to control timing of the Hsp90 reaction cycle[13] that can be targeted by co-chaperones and/or post translational modifications. For example, efficient maturation of CFTR requires an extended time bound to Hsp90, which can be achieved by an increased interaction of p23/Sba1 that prolongs the ATP-bound state, or reduced interaction with Aha1 that accelerates ATP hydrolysis[55,56]. Regulating advancement to the ADP-bound state by such effects on ATP hydrolysis can provide ways to fine-tune the Hsp90 cycle for specific clients and to broaden the range of its activities to fit the specific needs of its many clients.

For a long time, it has been thought that Hsp90 requires ATP hydrolysis to function. Many studies have interpreted data through that lens. Our work shows that nucleotide exchange is sufficient for Hsp90 function in vivo and may serve as the basis for reinterpretation of some past findings. Understanding the role of nucleotide exchange in Hsp90 function and how it relates to hydrolysis might form a foundation for new discoveries regarding Hsp90 activities and perhaps reveal new targets for Hsp90-specific therapies. For example, the EA substitution inhibited hHsp90β and EhHsp90 but improved hHsp90α and PfHsp90. Understanding these ortholog-specific effects may provide valuable insight in designing paralog- or species-specific compounds.

## Methods

### Yeast media, strains and plasmids

*S. cerevisiae* media for non-selective growth was YPAD or YPD (1% yeast extract, 2% peptone, 2% glucose, with or without 0.01% adenine) and for selective growth synthetic complete (SC, 2% glucose, 0.67% yeast nitrogen base with ammonium sulfate, SC drop-out mix as appropriate (Sunrise Scientific)). SCSuc, used for growing cells in the *GAL* induction assay (see below) is identical to SC except it contains 2% sucrose instead of glucose. Solid medium contained 2% agar. When used, calcofluor white (Sigma F3397) was added to YPD plates at 5 mg/L. 5′-fluoro-orotic-acid (FOA, US Biological F5050) was used in SC dropout plates at a final concentration of 1 g/L. For non-selective growth of *Sz. pombe* we used YES (3% glucose, 0.5% yeast extract, 0.0225% each: adenine, histidine, leucine, lysine, uracil) and for selective growth SC or pombe medium glutamate (PMG, Sunrise Scientific 2060-500) supplemented with appropriate nutrients. All cells were grown at 30 °C unless otherwise indicated.

Yeast strains used in this study are listed in Table 1. All *S. cerevisiae* strains are isogenic to BY4741[57]. The wild type parental *S. cerevisiae* strain used in plasmid shuffles, MR1075, has been described previously[33]. Strains MR1112 and MR1123 are isogenic to MR1075 except they are also *slt2*::KanMX or *PKC1-GFP*::His3MX, respectively. They were created using standard yeast genetic methods[33].

For plasmid shuffles in *Sz. pombe*, we made strain MRP1. First, we obtained the commercially available *Sz. pombe* heterozygous *hsp90*::KanMX6 diploid (Bioneer SPAC926.04c) and transformed it with plasmids pHL2806, a *leu1*⁺-marked plasmid required for *Sz. pombe* diploid sporulation[58], and pMR477, a *ura4*⁺-marked plasmid encoding *hsp90*⁺ (see Table 2). Pooled transformants from 5 colonies were suspended in sporulation medium (2% glucose, 0.17% yeast nitrogen base without ammonium sulfate, 0.1% glutamate, 0.02% adenine) and incubated with rolling at 25 °C for 3 days. Cells were pelleted by centrifugation, washed and resuspended in 2 mL water to OD$_{600}$ ~ 0.5. Next, 20 μL of a 1:20 dilution of glusulase (Perkin Elmer NEE154001EA) was added to digest asci and unsporulated diploids and incubated with rolling at 30 °C for 6 h. Cells from 1 mL of suspension were collected by centrifugation and resuspended in 300 μL of 30% ethanol. After incubating on the bench for 30 min, the cells were pelleted, resuspended in 100 μL water and spread onto solid YES medium containing 100 mg/L G418 (A. G. Scientific G-1030). The plate was incubated at 32 °C for 4 days. Twenty G418 resistant colonies were patched to a YES plate, and after incubation at 32 °C overnight this plate was replica-plated to: 1. YES containing 1 g/L FOA to select for genomic knock out of *hsp90* covered by pMR477; 2. YES lacking additional adenine to distinguish between haploids (red) and diploids (white); 3. PMG lacking leucine to identify isolates that lost plasmid pHL2806. Out of the FOA sensitive haploid isolates (19/20), the one that had the least background growth on FOA and was also leu⁻ was selected and named MRP1. The MRP1 genotype was verified by PCR and validated by ensuring that it became FOA resistant when transformed with plasmid pMR497 (*hsp90*⁺/*leu1*⁺; see Table 2) but not an empty vector.

Plasmids used are listed in Table 2. Plasmids pMR325, pMR423 and pMR365, for expression of Hsp82, hHsp90α or hHsp90β in *S.*

*cerevisiae* under the control of the strong GPD promoter, respectively, have been described[33,59]. Plasmid pMR55W is pRS314[60] with *HSC82* containing 500 bp of up- and down-stream sequence on a *Bam*HI fragment. Plasmids pMR453, pMR454 and pMR455, for expressing the Hsp90 orthologs from *C. neoformans*, *E. histolytica* or *P. falciparum* in *S. cerevisiae* under the control of the GPD promoter, respectively, were made by first having the open reading frames synthesized to add a *Spe*I site upstream of the ATG and a *Xho*I site downstream of the stop codon (Genewiz custom synthesis, sequences available upon request). These were then cloned into p414-GPD[61] cut with the same restriction enzymes. pKG2, for expressing *Sz. pombe* Hsp90 in *S. cerevisiae*, was made similarly except the open reading frame was amplified via PCR using *Sz. pombe* genomic DNA as template. The $P_{GAL10}$-driven luciferase reporter plasmid pMR169 has been described[62]. Plasmid pHL2806, which encodes genetic information necessary for *Sz. pombe* h⁺/h⁺ diploids to sporulate (a kind gift of Henry Levin), was described previously[58]. Plasmid pMR477 is the *Sz. pombe hsp90*⁺ gene with 669 bp of up- and 316 bp of downstream sequence on a *Bam*HI fragment cloned into plasmid pUR18[63]. Plasmid pMR496 is an empty vector containing the promoter of the *Sz. pombe tdh1*⁺ gene (analogous to the *S. cerevisiae* GPD promoter), a multiple cloning site (MCS), and the terminator from the *S. cerevisiae CYC1* gene. It was made by first amplifying 1000 bp upstream of the *tdh1*⁺ gene via PCR to add flanking *Sac*I and *Spe*I sites. The digested product was then cloned into plasmid pSP1[64] to give pSP1-P*tdh1*. Into this plasmid was then inserted a custom synthesized fragment containing *Sal*I-*Xho*I-T$_{CYC1}$-*Apa*I (sequence available on request) to give plasmid pMR496. Plasmids pMR497, pMR498, pMR499, pMR500 and pMR501 are the SpHsp90, Hsp82, Hsc82, hHsp90α or hHsp90β open reading frames, respectively, cloned into pMR496 as *Spe*I/*Xho*I. All point mutations were made using the Quickchange Lightning Multi site directed mutagenesis kit (Agilent 210515-5). All oligonucleotides used in this study are listed in Supplementary Table 1.

### Plasmid shuffling

Plasmid shuffling in *S. cerevisiae* was performed by first transforming MR1075 with *TRP1* plasmids (see Table 2) using a standard lithium acetate protocol. At least three Trp⁺ transformants for each construct, including empty vector and positive controls, were picked to a master SC -Trp plate. After overnight incubation at 30 °C, the master plate was replicated to identical media but containing 1 g/L FOA. Viability was scored after incubation at 30 °C for 3 days[33,59]. Images in the figures were created from pools of the primary transformants following verification of isolate phenotype consistency.

Plasmid shuffling in *Sz. pombe* was similar. *Sz. pombe* strain MRP1 was transformed with *leu1*⁺-marked plasmids using a lithium acetate method[65]. Transformants were selected on PMG or SC plates lacking leucine. At least four transformants from each transformation reaction were patched onto a PMG -leucine master plate and incubated at 30 °C overnight. Isolated *Sz. pombe* transformants were not pooled as was done for *S. cerevisiae* when preparing final images, as isolate heterogeneity was higher in *Sz. pombe* than *S. cerevisiae*. The master plate was then replica-plated to SC -leucine containing 1 g/L FOA. In our hands, background growth of Ura⁺ *Sz. pombe* on FOA was much reduced on SC medium compared to PMG. Plates were scanned after 4 days of incubation at 30 °C.

### Growth curves

FOA resistant *S. cerevisiae* cells were grown overnight in YPAD at 30 °C with rolling. One μL of overnight culture (or a volume to equalize cell densities across samples) was diluted into 150 μL of YPAD, or with radicicol (Cayman Chemical 13089) as indicated, in a 96 well plate that was incubated at 30 °C with orbital shaking (200 rpm) for 24 h. in a BMG Omega plate reader using BMG software version 5.5. The OD$_{600}$ was measured every 20 min. Three biological replicates derived from

individual transformation events were used for each sample. Doubling times were calculated from the steepest 4 h. portion of the linear growth phase.

## Calculation of mutant Hsp90's relative efficiency

To estimate the relative efficiency of the mutated Hsp82 or Hsc82 proteins, we used the relationship between growth rate and expression level of wild type Hsp82 as determined by the Bolon lab[38]: relative efficiency = (0.014*y)/(1-y) where y is the relative growth rate of the mutant compared to wild type. This value was normalized by dividing by the fold increase in expression of the mutant, as determined by analysis of western blots.

## Protein purification

His$_6$-tagged Hsp82 and Aha1 proteins were expressed in Rosetta 2 (DE3) *E. coli* transformed with plasmids pSK59 or pSK92, respectively[33,59]. Cells were lysed with a French press then clarified via centrifugation at 12,500 × *g*. His$_6$-tagged Hsp82 proteins were purified with immobilized metal affinity chromatography (IMAC) and gel filtration. His$_6$-tagged Aha1 was purified by IMAC, then the His$_6$-tag was removed via digestion with thrombin. Proteins were then reapplied to IMAC resin to remove uncut His6-Aha1[33,59]. All steps except the thrombin digestion were done at 4°C and/or on ice.

## PET and FRET measurements

For PET experiments, 10 μM of Hsp82 with the appropriate engineered cysteine/tryptophan pair[7] and with or without E33A as indicated was labeled with atto-Oxa11 maleimide (AttoTEC AD-oxa11-41) at a final concentration of 13 μM with 0.1 mM TCEP at room temperature for 2 h[7,33]. Excess dye was removed by spin desalting columns. Reactions were done in black 96 well plates with a 100 μL reaction volume with 2 μM unlabeled proteins (without the engineered cysteine and tryptophan) mixed with the matching 0.1 μM labeled protein in 25 mM HEPES pH 7.3, 200 mM NaCl and 5 mM MgCl$_2$. Fluorescence intensity (620 nm excitation, 680 nm emission) was measured every 20 s.

For FRET, 10 μM of purified Hsp82 proteins with either Q385C or D61C and E33A as indicated were labeled with either atto-488 maleimide (AttoTEC AD-488-41, on Q385C) or atto-550 maleimide (Atto-TEC AD-550-41, on D61C), respectively, at a final concentration of 13 μM with 0.1 mM TCEP at room temperature for 2 h. Excess dye was removed by spin desalting columns. Reactions were done in black 96 well plates with a 100 μL reaction volume and approximately 200 nM of each labeled protein in 25 mM HEPES pH 7.3, 200 mM NaCl and 5 mM MgCl$_2$. Mixtures were excited with 485 nm light and emission at 590 nm was measured every 20 s.

For both PET and FRET experiments, AMP-PNP (Roche 10102547001), ATP-γ-S (EMD Millipore 119120) or ATP (Roche 11140965001) as indicated were injected after ten minutes of baseline equilibration to a final concentration of 2 mM. Data were collected in a BMG Omega plate reader using BMG software version 5.5. Shown is the raw data and the exponential decay fits of the average of at least three independent experiments, using Prism 9 (GraphPad).

## Spot assays

FOA resistant *S. cerevisiae* or *Sz. pombe* cells were grown overnight in the appropriate non-selective liquid media at 30 °C with rolling. The OD$_{600}$ was adjusted to 0.25 and serially diluted (1:5) with sterile water. Four μL of each dilution was spotted onto non-selective solid media as indicated. *S. cerevisiae* plates were scanned after 2 days and *Sz. pombe* plates were scanned after 3 days of incubation at the indicated temperatures.

## BCIP assay

Defects in *S. cerevisiae* cell walls were detected by spreading 5 mL of melted 1% agarose, cooled to ~45 °C, containing 50 mM glycine pH 9.5

and 10 mM 5-bromo-4-chloro-3-indolyl phosphate (BCIP, Sigma B6149) onto YPD plates that had been used in spot assays. Blue color was allowed to develop at room temperature for 10–20 min before imaging[66].

## GAL induction assay

*S. cerevisiae* strain MR1075 was co-transformed with plasmids expressing various Hsp90 orthologs as indicated and plasmid pMR169. Transformants were selected on SC lacking tryptophan and histidine. Following plasmid shuffle via FOA (see above), individual biological replicates (three for each sample) were inoculated into liquid SCSuc lacking tryptophan and histidine. After overnight incubation at 30 °C with rolling, the cultures were diluted back 1:5 in identical medium and incubated for approximately 4 h. to attain log-phase growth. Radicicol (60 μM) was added to duplicate wild type cultures one hour before the initial reading. The OD$_{600}$ of the cultures was recorded and baseline luciferase activity was measured by mixing 100 μL of culture with 50 μL of 1 mM beetle luciferin (Promega E1602) in 0.1 M sodium citrate pH 5.0 and immediately reading in a Zylux Femtomaster luminometer, with a 5 s. delay and 5 s. read time. Galactose was then added to the cultures (final concentration 2%) and the cultures were incubated at 30 °C for 4 h. with rolling. The OD$_{600}$ and induced luciferase activity were then measured as before. Relative light units (RLU) per OD$_{600}$ was calculated to correct for differences in growth over the course of the experiment.

## Pkc1-GFP localization

*S. cerevisiae* strain MR1123 was transformed with plasmids expressing Hsp82 or Hsp82-E33A. After plasmid shuffling, FOA-resistant isolates were grown overnight in liquid YPAD at 30 °C with rolling. The cultures were diluted back 1:5 and incubated as before to achieve log phase, approximately 4 h. Cells from an aliquot were pelleted, washed and resuspended with water. Cells were imaged using a Nikon Eclipse Ni microscope equipped with a Q-Imaging Retiga EXi digital camera, Plan APO VC 60X oil immersion differential interference contrast optics, and shortpass GFP filter.

## Western blotting

Lysates for western blots were prepared by resuspending 50 OD$_{600}$ units of cold water-washed cells from log-phase cultures in 500 μL of lysis buffer (25 mM HEPES pH 7.4, 150 mM NaCl, 2 mM MgCl$_2$, 0.1% Triton X-100 with protease inhibitors) and lysing via mechanical breakage with glass beads[33]. Lysates were cleared via centrifugation at 3000 × *g* for 5 min and a portion was immediately mixed with an equal volume of 2X SDS-PAGE loading dye (0.1 M Tris-HCl pH 6.8, 3% SDS, 10% glycerol, 1% β-mercaptoethanol, trace bromphenol blue). Total protein concentration was determined using the Bradford assay (Bio-Rad 5000205). All cell and protein handling steps were done at 4°C and/or on ice. Proteins were separated on 4–20% Tris-HCl Criterion gels (Bio-Rad 5671095) and transferred to polyvinylidene fluoride membranes. For detecting *S. cerevisiae* Hsc82 and Hsp82, we used anti-Hsc82 (Abcam ab30920); for detecting Slt2 protein we used anti-Mpk1 E-9 (Santa Cruz Biotech sc-133189); for detecting the phosphorylated form of Slt2 we used P-p44/42 MAPK (Cell Signaling Technology 4370 S). All antibodies were used at 1:2000 dilution.

## Second site suppressor screen

To find intragenic suppressors of hHspp90β-EA's slow growth phenotype, the *E. coli* mutator strain XL1-Red (Agilent 200129) was transformed with plasmid pMR365-E42A, and a library of random mutants was created according to the manufacturer's protocol. *S. cerevisiae* strain MR1075 was transformed with this mutant library and approximately 10,000 colonies were selected on SC lacking tryptophan and screened by replica-plating the primary transformation plates to identical medium but containing FOA. Fast growing FOA resistant colonies (compared to a non-mutagenized control transformation) were picked.

False positives were screened out in a secondary screen by incubating replicas of the isolates at 37 °C, assuming the suppressed hHsp90β-EA allele would remain temperature sensitive, since *S. cerevisiae* expressing wild type hHsp90β grows slowly at elevated temperature. False positives arising from inter-plasmid recombination between the parental *URA3* and mutant *TRP1* plasmids or from suppressors in *URA3* grew well at 37 °C and thus were excluded. Plasmids were recovered from the fast growing (at 30 °C), temperature sensitive (at 37 °C) isolates and sequenced to identify the mutations.

**ATPase measurements**
ATP hydrolysis rates were measured in 100 µL reactions containing 2 µM (monomer) of the indicated Hsp82 proteins and 25 mM HEPES pH 7.4, 7 mM NaCl, 5 mM MgCl$_2$, 1 mM DTT, 0.6 mM NADH$^+$, 10 mM phospho-enol-pyruvate, and 2.5% pyruvate kinase/lactate dehydrogenase (Sigma P0294)[33]. Where included, Aha1 was at 4 µM. Reactions were initiated by the addition of ATP to 2 mM. Background ATPase activity was controlled for by measuring ATPase of matched samples containing radicicol (200 µM) and subtracting these values from the non-inhibited rates. Data were collected in a BMG Omega plate reader using BMG software version 5.5.

**Reporting summary**
Further information on research design is available in the Nature Portfolio Reporting Summary linked to this article.

## Data availability
The authors declare that all data supporting the findings of this study are available within the paper and its supplementary information files. Images of protein structures were sourced from Protein Database files 2CG9 [https://doi.org/10.2210/pdb2CG9/pdb] and 5FWK [https://doi.org/10.2210/pdb5FWK/pdb]. Source data are provided with this paper.

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

## Acknowledgements

We thank our NIH colleagues for helpful discussions and critical reading of the manuscript. We are very grateful to Henry Levin (NICHD) for plasmids and advice for establishing our *Sz. pombe* system. This research was supported by the Intramural Research Program of the NIH, the National Institute of Diabetes and Digestive and Kidney Diseases (NIDDK), award number ZIA DK024946-24 (D.C.M.).

## Author contributions

M.R. conceived and performed experiments and wrote the manuscript. K.G. conceived and performed the genetic screen. D.C.M. conceived experiments and assisted in writing the manuscript.

## Funding

## Competing interests

The authors declare no competing interests.
