## [Peer Review File · Nature Communications]

Nucleotide exchange is sufficient for Hsp90 functions in vivoREVIEWER COMMENTS

Reviewer #1 (Remarks to the Author):

The manuscript by Reidy et al. reports investigations on the role of nucleotide binding and hydrolysis in chaperoning by the heat shock protein Hsp90. The authors infer a nucleotide exchange mechanism that contrasts and adds to the common hydrolysis mechanism. Reidy et al. use viability and growth assays of *S. cerevisiae* and *Sz. pombe* cells containing engineered plasmids for expression of Hsp90, its orthologs, and mutants thereof that mediate wild-type, nucleotide-binding and -hydrolysis defective chaperone behavior. They further screened, applied, and investigated second-site suppressor mutations that alleviate nucleotide hydrolysis deficient mutants. The authors apply in-vitro biophysical measurements to detect conformational changes of isolated Hsp90 and its mutants in response to nucleotide binding using photoinduced electron transfer (PET) and fluorescence resonance energy transfer (FRET) probes. Reidy et al. arrive at the conclusion of having identified a nucleotide exchange mechanism in the Hsp90 catalytic cycle, besides the established hydrolysis mechanism.

Much progress has been made in past years in revealing the structure-function relationship of Hsp90 and its mechanism of chaperoning. Still, the Hsp90 chaperone belongs to the least well understood within the family of heat shock proteins. Currently, ATPase activity is regarded as critical. However, Hsp90's ATPase activity is unusually low. Results presented by Reidy et al. conclusively show that parts of Hsp90's chaperone activities do not require ATP hydrolysis. This is an interesting finding, which, as the authors show, is conserved across orthologs. The manuscript is well written.

However, I have concerns and comments, in particular regarding the authors' main conclusion of a nucleotide exchange mechanism, which are detailed below.

There is a misinterpretation of PET and FRET fluorescence data of modified Hsp90 measured in response to nucleotide binding, shown in Figure 2 and analyzed and discussed on pages 6-7 and 11-13. The authors erroneously interpret the lack of amplitude (signal change) of the fluorescence decay measured for the wild-type construct in response to application of ATP, which contrasts with the large-amplitude decay measured in response to application of non-hydrolysable ATP analogues, as a lack of conformational change (Figures 2B and 3E). But the small amplitudes of the decays measured in response to ATP result from dynamic binding and hydrolysis of the nucleotide in steady state, which is possible for ATP but not for non-hydrolysable analogues. In the ATP experiment the Hsp90 molecules cycle stochastically through the closed- and open-clamp conformations, driven by nucleotide binding and hydrolysis. In this dynamic equilibrium there is always a mixture of open and closed conformations present in the large ensemble of Hsp90 molecules probed at a time, and thus the apparently small change of fluorescence signal. After the short initial phase of nucleotide binding, which is observable as a small PET decay or FRET rise, the system is in steady state where Hsp90 molecules bind and hydrolyze

ATP dynamically and continuously (dynamic and continuous closing and opening of the molecular clamp), exactly as predicted from classical enzyme kinetics. The lack of amplitude in the ATP experiments shown in Figs. 2B and 2E is thus not a result of a lack of conformational change. However, it is worth noting that the relative difference of decay amplitudes in the AMP-PNP and ATP time traces contains additional interesting information. It shows that the opening event of the Hsp90 clamp after hydrolysis is faster than the closing event after ATP-binding to the apo state. The equilibrium constant of clamp closing is $K = k_c/k_o$ (with k_c = the rate constant for closing and k_o = the rate constant for opening) and thus the small signal change measured in the steady-state dynamic equilibrium. The result is in reasonable agreement with microscopic rate constants of clamp closure and opening extracted from single-molecule fluorescence intensity time traces measured in the presence of excess ATP, showing that opening is about five-fold faster than closing (ref. 8). The biophysical data are difficult to rationalize by a nucleotide exchange model but rather provide evidence for nucleotide hydrolysis.

The finding of the authors that binding of non-hydrolysable ATP analogues to Hsp90 mutant E33A resulted in weak decay amplitudes, contrasting the measurements with ATP (Figs. 2C and 2F), is indeed puzzling, as pointed out by the authors. A possible explanation is that the mutation E33A changes electrostatics in the ATP binding pocket such that binding of non-hydrolysable ATP, which exhibits modulated electrostatics compared to ATP caused by replacement of a phosphate oxygen by nitrogen or sulfur, is no more possible.

The nucleotide exchange model depicted in Figure 7 is not reasonable. Structural studies show that in the closed-clamp conformation of Hsp90 a polypeptide segment called the "lid" is folded over the ATP binding pocket in the N-terminal domain, thus trapping the nucleotide. For nucleotide exchange to occur within the closed-clamp conformation (as depicted in Figure 7) solely the lid would need to remodel/unfold, a hypothesis that conflicts with cooperativity of conformational changes during the catalytic cycle (refs. 7 and 8). Also, ATP rearrangement within the nucleotide binding pocket, as proposed in Figure 7 and discussed in the text, is unlikely to happen considering the perfect fit of the nucleotide in the shape of the pocket seen in structural studies.

Despite these concerns, the authors made an important point showing that viability and growth of cells containing ATP hydrolysis-deficient Hsp90 is still possible. It appears that ATP binding and subsequent closure of the molecular clamp is sufficient for parts of Hsp90's chaperoning functions. Hydrolysis of ATP in Hsp90 restores the apo state and the open-clamp conformation. The deficiency of hydrolysis-deficient mutants in restoring the apo state may be compensated by high expression levels of Hsp90, which can explain viability without nucleotide exchange.

In summary, the experiments and analyses presented by authors conclusively show that chaperoning activity of Hsp90 is possible without ATP hydrolysis, but I cannot see evidence for a nucleotide exchange mechanism.

Reviewer #2 (Remarks to the Author):

Reidy et al address a long-standing question in the Hsp90 field, namely the function of ATP hydrolysis and nucleotide exchange. Therefore, they use Hsp90 mutants that can either only bind (but not hydrolyze) ATP or even not bind ATP at all. Then they investigate how these mutants affect Hsp90's conformational dynamics and viability under various conditions in *S. cerevisiae* and *Sz. Pombe*. Finally, to shed light on the mechanism, they identified second-site suppressors of the hydrolysis deficient mutant. Altogether, this extensive study supports that exchange of ATP for ADP is critical for Hsp90 function, but not ATP hydrolysis itself. ATP hydrolysis likely provides an important control point for Hsp90 in its response to regulation by co-chaperones or processing of clients.

This is a very important and timely study, which sheds light on how nucleotides affect Hsp90 function. Nevertheless, in my opinion, several points have to be clarified and more of the taken experimental data has to be shown before I can fully support publication in NatComm:

1. Most importantly, the authors often speak about Hsp90's "cycle", which implies directionality for many people, for myself and obviously also for the authors, as e.g. shown in their Fig. 7. But there cannot be any directionality in the absence of an external energy source (i.e. ATP hydrolysis). This has to be clarified throughout the manuscript and also in Fig. 7 (e.g. the arrows have to go in both directions there).
2. In the introduction the authors mention a "closed-2" state of Hsp90. This is not well defined. In fact, I cannot find any "closed-2" state in the references given (7,8,9). There is a "closed 2" state (also slightly differently named) given in these publications:
<https://www.pnas.org/doi/suppl/10.1073/pnas.1414073111> and
<https://www.pnas.org/doi/epdf/10.1073/pnas.1916030116> but none of these references is cited here, therefore I am not sure which state the authors refer to – please clarify.
3. In several places the authors state that ATP induces different conformations in wild type Hsp82 compared to non-hydrolysable analogs and cite different literature in the different places. I think there should always be all references cited that show that (i.e. their references 14, 21,22).
4. For the FRET measurements: please state how the proteins were labeled (manufacturer's instructions sometimes change).
5. Fig. 2: Please show the raw data for the PET and FRET measurements and not only the fits, this is important to judge this data. At what time were the nucleotides added? This applies also to supplement Fig. 2.
6. Fig. 3D: I cannot see the localization to sites of polarized growth and bud necks. Maybe you can overlay the two images?

Reviewer #3 (Remarks to the Author):

The molecular chaperone Hsp90 is an ATP-dependent, essential protein in yeast. Using a hydrolysis-deficient Hsp90 mutant, the authors show that some non-essential but not the essential functions of Hsp90 in yeast depend on the ability of Hsp90 to hydrolyze ATP. Instead, their findings support the suggestion that exchange of ATP for ADP is critical for Hsp90's essential function in yeast. They also discovered a suppressor that could restore the interaction with clients and mitigate the mutation-induced phenotype. This is an interesting study that sheds new light on the molecular mechanism of Hsp90. While the concept presented is convincing, a few open issues need to be addressed.

Specific points

1. Controls regarding expression levels, especially for hHsp90, are required to rule out effects due to potentially different protein concentrations in vivo (Fig. 1 A).
2. Using PET and FRET setups, valuable insights into nucleotide-induced conformational changes could be obtained. Surprisingly, AMP-PNP does not lead to a completely closed state, which could hint towards a more selective binding of nucleotides by the mutant. The statement of an opposite effect of "the analogs" (only AMP-PNP) on the EA mutation does not seem to fit those results
3. Strictly speaking, the findings are limited to Hsp90 in *S. cerevisiae*. However, the authors claim that the viability of evolutionarily distant eukaryotic organisms does not depend on ATP hydrolysis (l. 26-27). Since the authors look exclusively at the survival of *S. cerevisiae* and *Sz. Pompe* (evolutionarily relatively close) expressing different Hsp90 orthologs, no conclusion can be drawn regarding the relevance of ATP hydrolysis for the survival of the donor organisms of the Hsp90 orthologs (e.g. human) as the authors also state in the discussion. In addition, the Hsp90 orthologs show some interesting differences.
4. The EATA and EAEK mutants are interesting regarding their ability to enhance ADP exchange. Here data for BCIP, sorbitol or growth at 37°C would be interesting.
5. The authors don't show the growth of the plasmid shuffle strains on plates without FOA. This information should be added (in the supplement).
6. Figure 2: Only the fit of the experiments is shown. Please include the raw data and the standard deviation.
7. Figure 3C: Which loading control is used? It looks like there are variations. The loading control is a uniform blot but the blot against Slt2 and P-Slt2 does not look like a uniform blot.

The authors need to discuss whether the decrease of phosphorylation can be a result of the overall decreased levels of Slt2.
8. Figure 3D: The background of Hsp82EA seems quite high. An assessment of whether a specific localization is present does not seem to be possible.
9. Figure 6: The hydrolysis rate of the Hsp82EAKA mutant is missing.

Reviewer #4 (Remarks to the Author):

The manuscript by Reidy, Garzillo, and Masison describes analyses of Hsp90 mutations that disrupt ATP hydrolysis in yeast. The authors assessed the E33A (EA) mutations in Hsp82 from budding yeast that disrupts a catalytic residue that positions a water for hydrolysis of ATP. They also examined mutations at the same catalytic position in Hsp90 from humans, fission yeast, and three other evolutionarily distant species. Using a plasmid swap approach they find that most EA variants were able to support sufficient growth of budding yeast under standard conditions to observe colonies on plates. In liquid culture, the authors observe that growth rate of yeast harboring EA mutations was impaired. The only Hsp90 with EA mutations that failed to support budding yeast growth was from fission yeast. The authors performed similar plasmid swap experiments in fission yeast and observed similar results – EA Hsp90 from fission yeast supported fission yeast growth into colonies under standard conditions, but budding yeast EA Hsp90 did not. The authors examined how EA Hsp90 impacted two environmental responses in budding yeast known to be Hsp90 dependent – cell wall integrity and gal induction. EA Hsp90 was defective in both cases. The authors went on to perform biophysical and biochemical analyses of Hsp90 proteins in purified form. Using both PET and FRET approaches, they observed that EA Hsp90 formed strongly compact closed conformations with ATP, but not AMPPNP or ATP γ S, whereas WT Hsp90 strongly closed with ATP analogues, but not ATP. The authors performed a screen for second mutations in EA Hsp90 that improved budding yeast growth. They performed follow up experiments on two of identified rescue variants – both located close to ATP in the structure of Hsp90. Both rescue variants showed partial rescue of growth rates in liquid culture and CWI and gal induction, indicating a general rescue of Hsp90 function. The authors propose a model where exchange of ATP for ADP, or ATP hydrolysis can perform similar steps in the Hsp90 cycle.

This work is of high significance because Hsp90 is a critical chaperone important for many signaling pathways and understanding its mechanism (the focus of this work) is relevant across wide fields of biology. The work has clearly been performed with proper attention to detail and the manuscript is reasonably straightforward to follow. I have a number of concerns that are listed below. I believe that these can be readily addressed by the authors, and would then make this work appropriate for publication in Nature Communications in my view.

Major points:

1. The authors reference prior work showing that Hsp90 protein level can be dramatically reduced without impacting yeast growth (ref 50). The authors should provide some estimation based on this or other work of how impaired the EA variants might be in terms of function relative to WT. For example, if slowing yeast growth 2-fold required 100x less Hsp90 (as in ref 50), then if the authors see 2-fold slower growth supported by EA that would be expected to roughly mimic a 100-fold decrease in function per molecule. I don't think that this takes away in any way from the importance of the authors findings, but it helps put them into evolutionary perspective (e.g. why they may not be observed in nature).
2. Related to point 1 above, are ATPase deficient Hsp90 mutations observed in the sequence databases? Either E33A or others? The authors should comment on this and how it relates to their observations.

3. The authors aptly point out that the different impacts of EA mutations across budding and fission yeast indicates some unknown functional divergence between Hsp90 in these two species. It would help the manuscript if the authors discussed the level of natural divergence between budding and fission yeast. Many readers might think that these species are closely related because they are both yeast. However most evolutionary models indicate that they are separated by almost as much evolutionary time as humans and either yeast. This makes the potential divergence between budding and fission yeast less surprising, especially when accounting for the faster generation time of yeast compared to humans.

4. The authors model of nucleotide exchange is appropriate and interesting. I would request that they clarify if they are distinguishing between exchange where binding and unbinding occur simultaneously, or if binding and unbinding are independent.

5. The authors nucleotide exchange model suggests that EA suppressor mutations would increase exchange rates. While it may be beyond the feasibility for this study, it would strengthen this work and the exchange mechanism conclusions if this could be experimentally tested for the two suppressor mutations studied in detail.

6. How common were EA suppressors? How many fast growing colonies to total colonies? Did the authors recover any A33E direct revertants?

7. Line 349 “While we cannot at this time explain the mechanism underlying these observations, 350 it is obvious that EA alters the conformational response to the different nucleotides, and that 351 non-hydrolysable analogs do not mimic the effect of ATP on wild type Hsp90’s movements.” I am not sure that this is true. Is there evidence that rules out transient occupation of an ATP bound state by WT that is distinct from the AMPPNP bound state?

Response to reviewers:

We thank the reviewers for their time and insightful comments on our manuscript. We have updated the text of the manuscript (in blue in the revised version), included new data and amended the references where appropriate. The Discussion has undergone significant revisions to address the reviewers' concerns and clarify our conclusions. We have changed the way some of the liquid culture growth data (Figures 1C, 1D, 4D and 5A, Supplemental Figure 1C) is presented in the revised version to doubling times of three biological replicates rather than change in OD over time – it is the same data as in the initial submission but presented differently in order to be more consistent. Our point-by-point responses to the reviewers' concerns are below in *italics*.

Reviewer #1 (Remarks to the Author):

The manuscript by Reidy et al. reports investigations on the role of nucleotide binding and hydrolysis in chaperoning by the heat shock protein Hsp90. The authors infer a nucleotide exchange mechanism that contrasts and adds to the common hydrolysis mechanism. Reidy et al. use viability and growth assays of *S. cerevisiae* and *Sz. pombe* cells containing engineered plasmids for expression of Hsp90, its orthologs, and mutants thereof that mediate wild-type, nucleotide-binding and -hydrolysis defective chaperone behavior. They further screened, applied, and investigated second-site suppressor mutations that alleviate nucleotide hydrolysis deficient mutants. The authors apply in-vitro biophysical measurements to detect conformational changes of isolated Hsp90 and its mutants in response to nucleotide binding using photoinduced electron transfer (PET) and fluorescence resonance energy transfer (FRET) probes. Reidy et al. arrive at the conclusion of having identified a nucleotide exchange mechanism in the Hsp90 catalytic cycle, besides the established hydrolysis mechanism.

Much progress has been made in past years in revealing the structure-function relationship of Hsp90 and its mechanism of chaperoning. Still, the Hsp90 chaperone belongs to the least well understood within the family of heat shock proteins. Currently, ATPase activity is regarded as critical. However, Hsp90's ATPase activity is unusually low. Results presented by Reidy et al. conclusively show that parts of Hsp90's chaperone activities do not require ATP hydrolysis. This is an interesting finding, which, as the authors show, is conserved across orthologs. The manuscript is well written.

However, I have concerns and comments, in particular regarding the authors' main conclusion of a nucleotide exchange mechanism, which are detailed below.

There is a misinterpretation of PET and FRET fluorescence data of modified Hsp90 measured in response to nucleotide binding, shown in Figure 2 and analyzed and discussed on pages 6-7 and 11-13. The authors erroneously

interpret the lack of amplitude (signal change) of the fluorescence decay measured for the wild-type construct in response to application of ATP, which contrasts with the large-amplitude decay measured in response to application of non-hydrolysable ATP analogues, as a lack of conformational change (Figures 2B and 3E). But the small amplitudes of the decays measured in response to ATP result from dynamic binding and hydrolysis of the nucleotide in steady state, which is possible for ATP but not for non-hydrolysable analogues. In the ATP experiment the Hsp90 molecules cycle stochastically through the closed- and open-clamp conformations, driven by nucleotide binding and hydrolysis. In this dynamic equilibrium there is always a mixture of open and closed conformations present in the large ensemble of Hsp90 molecules probed at a time, and thus the apparently small change of fluorescence signal. After the short initial phase of nucleotide binding, which is observable as a small PET decay or FRET rise, the system is in steady state where Hsp90 molecules bind and hydrolyze ATP dynamically and continuously (dynamic and continuous closing and opening of the molecular clamp), exactly as predicted from classical enzyme kinetics. The lack of amplitude in the ATP experiments shown in Figs. 2B and 2E is thus not a result of a lack of conformational change.

However, it is worth noting that the relative difference of decay amplitudes in the AMP-PNP and ATP time traces contains additional interesting information. It shows that the opening event of the Hsp90 clamp after hydrolysis is faster than the closing event after ATP-binding to the apo state. The equilibrium constant of clamp closing is $K = k_c/k_o$ (with k_c = the rate constant for closing and k_o = the rate constant for opening) and thus the small signal change measured in the steady-state dynamic equilibrium. The result is in reasonable agreement with microscopic rate constants of clamp closure and opening extracted from single-molecule fluorescence intensity time traces measured in the presence of excess ATP, showing that opening is about five-fold faster than closing (ref. 8).

1.

Reviewer #1 states that the PET & FRET experiments with wild type Hsp82 and ATP do not report on the conformation of Hsp82. This argument rests solely on presumed nucleotide binding dynamics driven by ATP hydrolysis. However, in the time course of these experiments, 20 min (in the presence of nucleotide), there is little to no measurable ATP hydrolysis occurring, since, as Reviewer #1 points out above, "Hsp90's ATPase activity is unusually low."

Considering this fact, if ATP binding alone did indeed cause Hsp90 to form the closed-clamp conformation, then one would observe rapid quenching of fluorescence in PET or increasing FRET upon addition of ATP (as the molecules bind to the nucleotide - which is in 1000X excess - and rapidly close), followed by a slower return of fluorescence in PET or loss of FRET signal as the Hsp90 molecules start to hydrolyze ATP and reopen. A new baseline would then form once equilibrium is established. As ADP accumulated, the ATPase rate would slow even further (since ADP inhibits the hydrolysis reaction), shifting the baseline even more. But this is not what

is observed. Furthermore, Hsp82-EA exhibits a very similar small PET decay/FRET increase upon addition of AMP-PNP that wild type Hsp82 does upon ATP injection (as pointed out below by Reviewer #1). This observation cannot be explained by Reviewer #1's model of hydrolysis-driven binding dynamics, as both E33A and AMPPNP preclude hydrolysis.

Taking together the PET/FRET data and the fact that ATPase is so slow, one can argue that the formation of the closed-clamp conformation, which is required for hydrolysis to occur, is a rare event in the presence of ATP. Indeed, the small dip in amplitude in wild type/ATP PET most likely reflects the small proportion of molecules that form the closed clamp. Instead, all our data fit the most parsimonious explanation, that ATP does not induce the closed-clamp conformation like AMP-PNP or ATP γ S do. E33A "flips" the conformational response to the various nucleotides, we think because of electrostatic changes to the nucleotide binding pocket that either do not allow ATP to adopt the correct conformation to induce closing or alter the ability of the protein to accurately sense the conformation of ATP.

We note that the other reviewers, #2 ("Then they investigate how these mutants affect Hsp90's conformational dynamics"), #3 ("Using PET and FRET setups, valuable insights into nucleotide-induced conformational changes could be obtained.") and #4 ("Using both PET and FRET approaches, they observed that EA Hsp90 formed strongly compact closed conformations...") all appreciate our use of PET and FRET experiments to understand the effects of E33A and/or different nucleotides on the conformational dynamics of Hsp90. Perhaps Reviewer #1 can concur that our conclusion is at least possible.

Aside from this point, Reviewer #1's comment brings to our attention that in a single experiment in Schubert et al. (initial submission and revision reference 8), the authors used ATP in a PET experiment. We have modified the second sentence in the second section of the Results to reflect this (revision lines 180-1).

The biophysical data are difficult to rationalize by a nucleotide exchange model but rather provide evidence for nucleotide hydrolysis.

2.

The PET and FRET experiments did not test the exchange mechanism, nor were they presented as such. Rather, we used the biophysical data to support the idea, presented here and before (e.g. Schmid & Hugel), that the orientation or conformation of ATP, specifically the γ phosphate, and not merely ATP binding per se, induces the closed-clamp conformation.

The finding of the authors that binding of non-hydrolysable ATP analogues to Hsp90 mutant E33A resulted in weak decay amplitudes, contrasting the measurements with ATP (Figs. 2C and 2F), is indeed puzzling, as pointed out by the authors. A possible explanation is that the mutation E33A changes electrostatics in the ATP binding pocket such that binding of non-hydrolysable ATP, which exhibits modulated electrostatics compared to ATP caused by replacement of a phosphate oxygen by nitrogen or sulfur, is no more possible.

3.

We do not think that E33A prevents binding of the non-hydrolysable analogs, since addition of them to E33A results in the "small" dip in PET/increase in FRET that is very similar to the effect of ATP on wild type. It is not suggested, for example, that the failure of ATP to produce large PET decay in wild type is due to wild type Hsp82 being unable to bind ATP.

Nevertheless, we do not have direct evidence that E33A affects binding of the non-hydrolysable analogs. We also were unable to find in the peer-reviewed literature a study that measured interactions of AMPPNP or ATP γ S with E33A. However, a Ph.D. dissertation from the Buchner lab (<https://mediatum.ub.tum.de/doc/1352918/1352918.pdf>) found E33A had little effect on binding of a fluorescently labeled version of AMPPNP. Since this concern was also raised by Reviewer #3, we have added a caveat to this effect in the Discussion (revision lines 369-71).

We do agree with Reviewer #1 that one possibility for how E33A “flips” the response to ATP vs. AMPPNP/ATP γ S, as described above and in the Discussion, is due to “modulated electrostatics” caused by the loss of negative charge in the binding pocket. In fact, we made a similar argument on lines 355-358 of the initial submission. However, we did not consider the different electrostatics of the analogs compared to ATP, and how the interplay between them and either E or A at 33 influences the conformational dynamics. We have updated that section of the Discussion (revision lines 382-3) to incorporate this idea and we thank the reviewer for this suggestion.

The nucleotide exchange model depicted in Figure 7 is not reasonable. Structural studies show that in the closed-clamp conformation of Hsp90 a polypeptide segment called the “lid” is folded over the ATP binding pocket in the N-terminal domain, thus trapping the nucleotide. For nucleotide exchange to occur within the closed-clamp conformation (as depicted in Figure 7) solely the lid would need to remodel/unfold, a hypothesis that conflicts with cooperativity of conformational changes during the catalytic cycle (refs. 7 and 8).

4.

While it is certainly true that the lid structure is positioned over the nucleotide binding site in the AMPPNP bound (“closed-clamp” or “closed-2”) crystal structure in Ali et al., careful inspection of this structure (pdb 2cg9) in surface view reveals the presence of a hole and a channel in which the entire adenosine ring of the nucleotide can be readily observed. Residues K44, I82, E88, N92, K102 and A103 comprise the outer rim of this channel, and the bound Sba1 does not block it. In solution the diameter of the channel is likely fluctuating as the protein “breathes”. Thus, the nucleotide is not trapped completely by the lid and there is a possible “escape hatch” through which it could dissociate. We have updated the Discussion to include this possibility (revision lines 419-21) and have added Supplemental Figure 5 (new to the revision) showing the channel in the Ali et al. structure.

Also, ATP rearrangement within the nucleotide binding pocket, as proposed in Figure 7 and discussed in the text, is unlikely to happen considering the perfect fit of the nucleotide in the shape of the pocket seen in structural studies.

5.

Early structural studies of the Hsp90 N-terminal domain were unable to see the γ phosphate of the bound ATP due to its movement (pdb 1am1, initial and revision ref 24). More recent work found that the closed-clamp structure is favored when the movement of ATP within the pocket is restricted (ref 37 of the initial submission, revision ref 40). These observations are sufficient to convincingly argue against the idea that ATP, specifically the γ phosphate, is static within the nucleotide binding pocket. Also, the closed Hsp82-Sba1 complex structure (pdb 2cg9) was made using AMPPNP and a mutant Hsp82 that had A107N and a shortened linker region, according to the Methods of Ali et al., even though it is almost universally referred to as the

“ATP state” of wild type Hsp82 (in the pdb file in the protein database the sequence is wild type at 107 and the nucleotide is ATP). More recent cryo-EM structures of closed hHsp90 β in complex with Cdc37 and kinase (pdb 5fwk, initial ref 48, revision ref 51) used the “ATP” from 2cg9 to model the nucleotide in their complexes which were isolated from Sf9 cells (i.e. with ATP). It is our conclusion that the conformation observed in 2cg9 is the **rearranged** conformation of ATP, that is, the conformation of ATP that induces closure and is mimicked by AMPPNP and ATP γ S. Thus, it makes perfect sense to interpret that the nucleotide adopts a “perfect fit” within the pocket of this structure. Since ATP binding per se is not enough to induce N-M docking or dimer closure, we hypothesize that, after binding, the ATP molecule must adopt the proper conformation for closing to occur, and that this conformation is the same one that AMPPNP assumes naturally. One intriguing and as yet untested possibility is that rather than trapping ATP, the lid structure (see above) influences its proper orientation.

Despite these concerns, the authors made an important point showing that viability and growth of cells containing ATP hydrolysis-deficient Hsp90 is still possible. It appears that ATP binding and subsequent closure of the molecular clamp is sufficient for parts of Hsp90’s chaperoning functions. Hydrolysis of ATP in Hsp90 restores the apo state and the open-clamp conformation. The deficiency of hydrolysis-deficient mutants in restoring the apo state may be compensated by high expression levels of Hsp90, which can explain viability without nucleotide exchange.

6.

Our work and the work of others (e.g. Lopez, ref 14) argues against the existence of an appreciably populated apo state in vivo. We made this point on lines 381-384 of the initial submission. Cellular concentrations of ADP and ATP are in the millimolar range, while the affinity for nucleotides is in the micromolar range. Thus, the nucleotide binding sites of Hsp90 are most likely occupied at any given moment. Our model is that the changes in the conformation from open (ADP) to closed (ATP) and back to ADP are essential for Hsp90 function but that the transition from ATP to ADP need not require hydrolysis. We have revised Figure 7 (in response to this criticism and other reviewers’ concerns) and the Discussion (revision lines 403-18) to make this point more clear.*

Also, the near wild type growth rate and lack of CWI defects of cells expressing Hsp82-EAEK (Hsp82-EA with the E384K suppressor) argues very strongly that ATP hydrolysis is dispensable for virtually all, not just “parts of Hsp90’s chaperoning functions”.

In summary, the experiments and analyses presented by authors conclusively show that chaperoning activity of Hsp90 is possible without ATP hydrolysis, but I cannot see evidence for a nucleotide exchange mechanism.

7.

Again, nucleotide exchange is the most parsimonious mechanism that explains all of our data. We note that the other reviewers agree that our work supports a model of Hsp90 function that incorporates nucleotide exchange. Reviewer #2: “...this extensive study supports that exchange of ATP for ADP is critical for Hsp90 function, but not ATP hydrolysis itself.” Reviewer #3: “...their findings support the suggestion that exchange of ATP for ADP is critical for Hsp90’s essential function in yeast.... the concept presented is convincing...”. Reviewer #4: “The

authors propose a model where exchange of ATP for ADP, or ATP hydrolysis can perform similar steps in the Hsp90 cycle...This work is of high significance..."

Furthermore, our model agrees with the work of many others, including but not limited to Zierer, et al., Halpin & Street, Schmid & Hugel, etc. We show conclusively that ATP hydrolysis is not required for either essential or non-essential functions and that progression back to the ADP state is necessary in the context of no hydrolysis (K98A + E33A is nearly lethal). We cannot envision another mechanism that can adequately and simply explain all of the data.

Reviewer #2 (Remarks to the Author):

Reidy et al address a long-standing question in the Hsp90 field, namely the function of ATP hydrolysis and nucleotide exchange. Therefore, they use Hsp90 mutants that can either only bind (but not hydrolyze) ATP or even not bind ATP at all. Then they investigate how these mutants affect Hsp90's conformational dynamics and viability under various conditions in *S. cerevisiae* and *Sz. Pombe*. Finally, to shed light on the mechanism, they identified second-site suppressors of the hydrolysis deficient mutant. Altogether, this extensive study supports that exchange of ATP for ADP is critical for Hsp90 function, but not ATP hydrolysis itself. ATP hydrolysis likely provides an important control point for Hsp90 in its response to regulation by co-chaperones or processing of clients.

This is a very important and timely study, which sheds light on how nucleotides affect Hsp90 function. Nevertheless, in my opinion, several points have to be clarified and more of the taken experimental data has to be shown before I can fully support publication in NatComm:

1. Most importantly, the authors often speak about Hsp90's "cycle", which implies directionality for many people, for myself and obviously also for the authors, as e.g. shown in their Fig. 7. But there cannot be any directionality in the absence of an external energy source (i.e. ATP hydrolysis). This has to be clarified throughout the manuscript and also in Fig. 7 (e.g. the arrows have to go in both directions there).

We have simplified our model in an updated Figure 7 by removing the cartoon showing the closed ADP state. We have also added an arrow pointing in the opposite direction between the open-ADP cartoon/open-ATP cartoon to reflect that these states exist in equilibrium. We have changed the arrow pointing towards the closed/ATP cartoon to a thin dotted line to reflect the biophysical data showing ATP binding alone does not efficiently induce closing and labeled this step "ATP γ phosphate repositioning". We speculate that in vivo closing with ATP involves client and co-chaperone interaction that provides directionality. As the reviewer points out, the hydrolysis step (or ATP-out/ADP-in exchange) also provides directionality, towards the ADP-open state (and the beginning of the cycle as defined here), so we have left that arrow pointing in one direction. The section of the Discussion dealing with Figure 7 has also been rewritten, in response to this and other reviewers' concerns (revision lines 403-18), and the legend for Figure 7 has been updated accordingly.

In light of Reviewer #2's comment, we see now that our use of the phrase "reaction cycle" was perhaps a little heavy. We have replaced the phrase "reaction cycle" in several places throughout the manuscript where appropriate, for example, where a more general point of Hsp90 function is being discussed, and not the "reaction cycle" specifically (revision lines 17, 70, 83-5, 87, 435, 437, 447, 450, 451).

We also point out here that the energy of nucleotide binding itself may be sufficient to drive conformation changes. We previously reported that Hsp82 in the closed-clamp conformation (in complex with AMPPNP) could be returned to the open state by adding ADP, via PET (initial and revision ref 33). This showed that hydrolysis is not necessary and exchange for ADP is sufficient for Hsp90 to open. Also, the energy from Hsp90 ATP hydrolysis is so low that it does not interfere with ITC experiments (revision ref 53), anecdotally suggesting that hydrolysis may not be the major way energy for movements is obtained.

2. In the introduction the authors mention a "closed-2" state of Hsp90. This is not well defined. In fact, I cannot find any "closed-2" state in the references given (7,8,9). There is a "closed 2" state (also slightly differently named) given in these publications: <https://www.pnas.org/doi/suppl/10.1073/pnas.1414073111> and <https://www.pnas.org/doi/epdf/10.1073/pnas.1916030116> but none of these references is cited here, therefore I am not sure which state the authors refer to – please clarify.

We were referring to the "closed-2" state as defined by Zierer et al. NSMB 2016, which as Reviewer #2 points out is not initial (nor revision) submission reference 7, 8 or 9. We apologize for the confusion caused by this oversight. Schulze, Schubert and Prodromou (refs 7,8,9 respectively) refer to the same conformation as "closed-clamp". We have updated the text throughout the manuscript where appropriate to refer to the closed conformation of Hsp90 as "closed-clamp" (revision lines 45, 171-3, 273, 412, 416, Figure 2 legend, Supplemental Figure 5 legend).

3. In several places the authors state that ATP induces different conformations in wild type Hsp82 compared to non-hydrolysable analogs and cite different literature in the different places. I think there should always be all references cited that show that (i.e. their references 14, 21,22).

We apologize for the lack of consistency on this point. We have revised the Introduction, Results and Discussion to include all of the following references that have reported differences in the effects of ATP vs. nonhydrolyzable analogs on Hsp90's conformation (revision lines 48-50, 185-6, 362-4):

- *Halpin & Street JMB 2016 (initial and revision ref 6): Figure 6A shows a FRET experiment where AMPPNP, but not ATP, induces closing of Hsp82.*
- *Zierer et al NSMB 2016 (initial and revision ref 13): Figures 4A shows ATP γ S inducing closing in wild type Hsp82 via FRET and Figure 4B shows ATP does not.*
- *Lopez et al Sci Adv 2020 (initial and revision ref 14): Figure 2A shows an NMR experiment where ATP does not induce Hsp82 closing as expected from analysis of 2cg9 (i.e., AMPPNP) but rather looks like apo.*
- *Mickler et al NSMB 2009 (initial ref 21, revision ref 15): using single molecule FRET the authors showed that Δ 8 Hsp82 with ATP behaved like wild type with AMPPNP.*

- *Giannoulis et al PNAS 2020 (initial ref 22, revision ref 16): the authors used AMPPNP to mimic the “prehydrolysis”, ie closed-clamp, state of Hsp82 in NMR experiments because ATP alone did not induce closing.*

Additionally, we cited Halpin, Zierer and Mickler in the Results where we report our FRET results with ATP (revision line 195), as these papers reported similar FRET data.

4. For the FRET measurements: please state how the proteins were labeled (manufacturer’s instructions sometimes change).

We have updated the Methods section to include these details (revision lines 558-60).

5. Fig. 2: Please show the raw data for the PET and FRET measurements and not only the fits, this is important to judge this data. At what time were the nucleotides added? This applies also to supplement Fig. 2.

Figure 2 and Supplementary Figure 2 have been updated to include the raw data. The nucleotides were added at $T = 10$, as described in the Methods section (line 520 of the initial submission). We have updated the legends of Figure 2 and Sup. Figure 2 to include this important information.

6. Fig. 3D: I cannot see the localization to sites of polarized growth and bud necks. Maybe you can overlay the two images?

The image of Hsp82-EA expressing cells in Figure 3D of the initial submission had high background which obscured the lack of enrichment of Pck1-GFP at actively growing areas and bud necks. This issue was also raised by Reviewer #3. We have readjusted the contrast of this image to normalize the background fluorescence of the two panels. We also added arrows to both fluorescent images in Figure 3D to point out the bud necks and sites of polarized growth.

Reviewer #3 (Remarks to the Author):

The molecular chaperone Hsp90 is an ATP-dependent, essential protein in yeast. Using a hydrolysis-deficient Hsp90 mutant, the authors show that some non-essential but not the essential functions of Hsp90 in yeast depend on the ability of Hsp90 to hydrolyze ATP. Instead, their findings support the suggestion that exchange of ATP for ADP is critical for Hsp90’s essential function in yeast. They also discovered a suppressor that could restore the interaction with clients and mitigate the mutation-induced phenotype. This is an interesting study that sheds new light on the molecular mechanism of Hsp90. While the concept presented is convincing, a few open issues need to be addressed.

Specific points

1. Controls regarding expression levels, especially for hHsp90, are required to rule out effects due to potentially different protein concentrations in vivo (Fig. 1 A).

Unfortunately, we have not been able to find an antibody that recognizes all the various isoforms of Hsp90 we study, despite testing many different commercially available anti-Hsp90 antibodies. For example, Abcam ab30920 recognizes *S. cerevisiae* Hsp82 and Hsc82 very well, human Hsp90 α and *Sz. pombe* Hsp90 extremely poorly and hHsp90 β not at all. Two antibodies from Enzo (ADI-SPS-771 or ADI-SPA-830) and one from CellSignalling Technologies (D1A7) recognizes hHsp90 α very well, hHsp90 β poorly but not Hsp82, Hsc82 or *Sz. pombe* Hsp90. StressMarq H9010 recognizes only hHsp90 β . Finally, a third Enzo antibody (ADI-SPA-840) recognizes only Hsp82 and hHsp90 α . None of the tested antibodies recognized Hsp90 from *C. neoformans* or *E. histolytica* or *P. falciparum*. The problem with performing the requested analysis is compounded further by the fact that cells expressing hHsp90 β -E42A grow so poorly that we were not able to collect enough for Western blots. Thus, due to technical limitations this important control cannot be performed, and we did not add tags to our constructs due to possible deleterious effects. Of all the non-*S. cerevisiae* Hsp90s, we could only evaluate hHsp90 α compared to hHsp90 α -E47A, and found they are expressed at similar levels.

Regardless, we have consistently observed that the levels of Hsp82 mutants and/or variants that cause slow growth phenotypes are actually higher than wild type Hsp90. Such is the case with Hsp82-EA and Hsc82-EA (Supplemental Figure 1A of the initial submission, Supplemental Figures 1B and 3B of the revision). One explanation of this would be that only cells that accumulate higher amounts of defective Hsp90s survive. Indeed, the expression levels of the rescued variants (EAEK) in either Hsp82 or Hsc82 are lower level than EA (revision Supplemental Figure 3B).

2. Using PET and FRET setups, valuable insights into nucleotide-induced conformational changes could be obtained. Surprisingly, AMP-PNP does not lead to a completely closed state, which could hint towards a more selective binding of nucleotides by the mutant. The statement of an opposite effect of "the analogs" (only AMP-PNP) on the EA mutation does not seem to fit those results

Reviewer #1 also raised this concern - that E33A selectively inhibits binding to the analogs. While we think this possibility is unlikely (see Reviewer #1, item #3 for our arguments), we have added this caveat to the Discussion (revision lines 369-71).

3. Strictly speaking, the findings are limited to Hsp90 in *S. cerevisiae*. However, the authors claim that the viability of evolutionarily distant eukaryotic organisms does not depend on ATP hydrolysis (l. 26-27). Since the authors look exclusively at the survival of *S. cerevisiae* and *Sz. Pompe* (evolutionarily relatively close) expressing different Hsp90 orthologs, no conclusion can be drawn regarding the relevance of ATP hydrolysis for the survival of the donor organisms of the Hsp90 orthologs (e.g. human) as the authors also state in the discussion. In addition, the Hsp90 orthologs show some interesting differences.

*We respectfully object to Reviewer #3's characterization of *S. cerevisiae* and *Sz. pombe* as being "evolutionarily relatively close". Even though *S. cerevisiae* and *Sz. pombe* are both yeasts they are quite distant from each other evolutionarily, on the order of 330-420 million years. In fact, this is the amount of evolutionary distance between either *S. cerevisiae* and humans or *Sz. pombe* and humans, as Reviewer #4 points out below (item #3). While we*

specifically noted that the two yeasts are “evolutionarily distant” in the Abstract (lines 26-27 of the initial submission), Introduction (line 103), Results (line 153) and mentioned the relative length of evolutionary distance in the Discussion (lines 337-338), we have updated the Results and Discussion to emphasize this point (revision lines 149-51, 300-2, 355-6) with added references (revision refs 36 & 37).

As Reviewer #3 points out, we state in the Discussion that “our work using fungal species does not address whether ATP hydrolysis is required for essential Hsp90 functions in metazoans...”. While it is currently unknown whether ATP hydrolysis is required for the viability of cells other than *S. cerevisiae* and *Sz. pombe*, the larger point we were making was that the Hsp90 **protein** does not need to hydrolyze ATP in order to keep eukaryotic cells alive and well, and that this property of the Hsp90 protein is conserved.

4. The EATA and EAEK mutants are interesting regarding their ability to enhance ADP exchange. Here data for BCIP, sorbitol or growth at 37°C would be interesting.

Sensitivity to calcofluor white (CFW), positive BCIP staining and osmo-remediated temperature sensitivity are all phenotypes that result from defects in the cell wall integrity (CWI) pathway (initial submission refs 38-41, revision refs 41-44 and reestablished here in Figure 3A). In Figure 5B we used CFW sensitivity as a representative CWI phenotype. We have included the requested data as Supplemental Figure 4A (new in the revision). As expected, EK rescued both the E33A-mediated osmo-remediated TS phenotype and BCIP staining, but TA only had a noticeable effect on BCIP staining, in line with other observations of EK mediating a stronger rescue effect. We have also added images showing the suppressive effect of TA and EK on CWI in cells expressing Hsc82-EA as Supplemental Figure 4B.

5. The authors don't show the growth of the plasmid shuffle strains on plates without FOA. This information should be added (in the supplement).

*We have added images of the FOA resistant (ie, plasmid-shuffled) *S. cerevisiae* strains from Figure 1A growing on YPD plates as revised Supplemental Figure 1A and those from Figure 4C as Supplemental Figure 3A (new to the revision).*

6. Figure 2: Only the fit of the experiments is shown. Please include the raw data and the standard deviation.

Figure 2 and Supplementary Figure 2 are now updated to include the raw data.

7. Figure 3C: Which loading control is used? It looks like there are variations. The loading control is a uniform blot but the blot against Slt2 and P-Slt2 does not look like a uniform blot.

The authors need to discuss whether the decrease of phosphorylation can be a result of the overall decreased levels of Slt2.

The loading control shown in Figure 3C is the amido black-stained membrane that was probed with anti-Slt2 antibody. Slt2 expression is feedback upregulated in response to Slt2

phosphorylation (ie, activation of the CWI pathway, by agents such as calcofluor white (CFW)), so both the levels of Slt2 protein (top panel in Figure 3C) and phosphorylated Slt2 (middle panel) are expected to go up in response to CFW treatment (right lanes "+ CFW") (initial ref 38, revision ref 41), as they do in cells expressing wild type Hsp82 ("P") or Hsc82 ("C"). The observation that Slt2 phosphorylation is much lower in cells expressing the EA version of Hsp82 ("p^{EA}") or Hsc82 ("c^{EA}"), and therefore the overall levels of Slt2 protein are lower, agrees with findings by others that Slt2 and other members of the CWI pathway are Hsp90 clients. We have updated the Results sections (revision lines 220-2) to provide clarification of this point.

8. Figure 3D: The background of Hsp82EA seems quite high. An assessment of whether a specific localization is present does not seem to be possible.

See our response to Reviewer #2, item #6. We agree that the background of this image in the initial submission Figure 3D was indeed high, and we have normalized the background intensity in the revision. We also have added arrows to point out the sites of polarized growth and bud necks to both images of Figure 3D.

9. Figure 6: The hydrolysis rate of the Hsp82EAKA mutant is missing.

The hydrolysis rate of Hsp82-EAKA (E33A+K98A) was not measured because the EA mutation abolishes the hydrolysis reaction regardless of secondary mutations, as was demonstrated for EATA and EAEK (Figure 6A). In fact, we did not purify this protein, as we only used EAKA to test the effect of reducing ADP affinity in the context of no hydrolysis on cell growth.

Reviewer #4 (Remarks to the Author):

The manuscript by Reidy, Garzillo, and Masison describes analyses of Hsp90 mutations that disrupt ATP hydrolysis in yeast. The authors assessed the E33A (EA) mutations in Hsp82 from budding yeast that disrupts a catalytic residue that positions a water for hydrolysis of ATP. They also examined mutations at the same catalytic position in Hsp90 from humans, fission yeast, and three other evolutionarily distant species. Using a plasmid swap approach they find that most EA variants were able to support sufficient growth of budding yeast under standard conditions to observe colonies on plates. In liquid culture, the authors observe that growth rate of yeast harboring EA mutations was impaired. The only Hsp90 with EA mutations that failed to support budding yeast growth was from fission yeast. The authors performed similar plasmid swap experiments in fission yeast and observed similar results – EA Hsp90 from fission yeast supported fission yeast growth into colonies under standard conditions, but budding yeast EA Hsp90 did not. The authors examined how EA Hsp90 impacted two environmental responses in budding yeast known to be Hsp90 dependent – cell wall integrity and gal induction. EA Hsp90 was defective in both cases. The authors went on to perform biophysical and biochemical analyses of Hsp90

proteins in purified form. Using both PET and FRET approaches, they observed that EA Hsp90 formed strongly compact closed conformations with ATP, but not AMPPNP or ATPgammaS, whereas WT Hsp90 strongly closed with ATP analogues, but not ATP. The authors performed a screen for second mutations in EA Hsp90 that improved budding yeast growth. They performed follow up experiments on two of identified rescue variants – both located close to ATP in the structure of Hsp90. Both rescue variants showed partial rescue of growth rates in liquid culture and CWI and gal induction, indicating a general rescue of Hsp90 function. The authors propose a model where exchange of ATP for ADP, or ATP hydrolysis can perform similar steps in the Hsp90 cycle.

This work is of high significance because Hsp90 is a critical chaperone important for many signaling pathways and understanding its mechanism (the focus of this work) is relevant across wide fields of biology. The work has clearly been performed with proper attention to detail and the manuscript is reasonably straightforward to follow. I have a number of concerns that are listed below. I believe that these can be readily addressed by the authors, and would then make this work appropriate for publication in Nature Communications in my view.

Major points:

1. The authors reference prior work showing that Hsp90 protein level can be dramatically reduced without impacting yeast growth (ref 50). The authors should provide some estimation based on this or other work of how impaired the EA variants might be in terms of function relative to WT. For example, if slowing yeast growth 2-fold required 100x less Hsp90 (as in ref 50), then if the authors see 2-fold slower growth supported by EA that would be expected to roughly mimic a 100-fold decrease in function per molecule. I don't think that this takes away in any way from the importance of the authors findings, but it helps put them into evolutionary perspective (e.g. why they may not be observed in nature).

We thank Reviewer #4 for this suggestion, as it provides an extremely informative way to quantify the effects of the mutants that we had not considered. We used the relationship between growth rate and expression level of wild type Hsp82 from Figure 2C of Jiang et al (initial submission ref 50, revision ref 38), and then normalized this to correct for the expression level of the mutants compared to wild type (using blots in initial submission Supplemental Figure 1A, revision Supplemental Figure 1B). We estimated the relative efficiency of Hsp82-E33A (1.8%), Hsc82-EA (0.8%), Hsp82-EATA (10%) and Hsp82-EAEK (~100%). We have updated the Results, Discussion and Methods (revision lines 161-4, 283-5, 427-8, 545-50) to include this analysis.

2. Related to point 1 above, are ATPase deficient Hsp90 mutations observed in the sequence databases? Either E33A or others? The authors should comment on this and how it relates to their observations.

We assume that Reviewer #4 is referring to the “sequence databases” of Mishra et al. Cell Rep. 2016 (initial submission ref 49, revision ref 52). We have updated the first paragraph of the Discussion (revision lines 341-50) to:

*“Here, we confirm that *S. cerevisiae* cells expressing ATP hydrolysis defective Hsp82 are viable. These findings agree with a recent report [13] but conflict with earlier papers [25, 26, 52]. We cannot explain the discrepancy, except to note that two earlier studies fused hexa-histidine tags onto their Hsp90s, which may have allele-specific effects [25, 26]. A more recent growth competition study among all possible mutations in the Hsp82 N-domain found every amino acid except glutamate at position 33 was “unfit” [52]. However, the mutants were expressed at a low level which reduces Hsp90 activity [38] and grown in a competitive environment, in contrast to our system where each mutant is tested for its individual ability to support growth. Importantly, our work and Mishra et al. together show that while ATP hydrolysis is not required, it is selected for, meaning that ATP hydrolysis provides an evolutionary advantage.”*

3. The authors aptly point out that the different impacts of EA mutations across budding and fission yeast indicates some unknown functional divergence between Hsp90 in these two species. It would help the manuscript if the authors discussed the level of natural divergence between budding and fission yeast. Many readers might think that these species are closely related because they are both yeast. However most evolutionary models indicate that they are separated by almost as much evolutionary time as humans and either yeast. This makes the potential divergence between budding and fission yeast less surprising, especially when accounting for the faster generation time of yeast compared to humans.

*We thank Reviewer #4 for the suggestion of more thoroughly discussing the level of divergence between *S. cerevisiae* and *Sz. pombe*. Reviewer #3’s characterization of the two yeasts as being “evolutionarily relatively close” (Reviewer #3, item #3) illustrates Reviewer #4’s concern. In the initial submission we noted in several places that the two species are “evolutionarily distant”, but we agree that the manuscript would benefit greatly from having more details regarding the length of evolutionary divergence. We updated the Results and Discussion to emphasize this point (revision lines 149-51, 300-2, 355-6) and included new references (revision refs 36 & 37).*

4. The authors model of nucleotide exchange is appropriate and interesting. I would request that they clarify if they are distinguishing between exchange where binding and unbinding occur simultaneously, or if binding and unbinding are independent.

Our data suggest that exchange happens more-or-less simultaneously. In the current model, ADP and phosphate are released after hydrolysis and Hsp90 assumes an apo state while it waits to rebind ATP. This model ignores the fact that there are mM concentrations of nucleotides in the cell and Hsp90’s affinity for ADP is higher than ATP by ~2.5 fold (according to a recent ITC study, revision ref 53). Thus, even though the ATP:ADP ratio in the cell is ~4:1 (revision ref 54), ADP likely remains bound after hydrolysis. Our data show that the conversion

from the ATP to ADP state need not rely on hydrolysis. Thus, our model states that the apo state of Hsp90 is rare enough to be effectively non-existent, and supports the idea that Hsp90 populates ADP/ADP, ATP/ADP and ATP/ATP states, as proposed by Halpin and others. It follows then that if there is no apo state, then a second exchange, from ADP to ATP, must occur. We speculate that due to the affinities for the nucleotides and their cellular concentrations, cellular conditions may exist on a “knife’s edge” where exchange in either direction can happen readily, or at least enough to work (we have not tested this idea directly yet). The high abundance of Hsp90 may facilitate favorable outcomes in this regard. We have updated the Discussion to clarify this point (revision lines 403-18, 421-5).

Our data do not shed much light on whether the hydrolysis and/or exchange of one protomer influences that of the other. This is a question we are interested in and are currently developing strategies to address.

5. The authors nucleotide exchange model suggests that EA suppressor mutations would increase exchange rates. While it may be beyond the feasibility for this study, it would strengthen this work and the exchange mechanism conclusions if this could be experimentally tested for the two suppressor mutations studied in detail.

We agree with Reviewer #4 that direct evidence of nucleotide exchange would add support to our conclusions. It is our understanding that the fluorescently labeled versions of ATP that are usually used for nucleotide exchange measurements are not bound by Hsp82, however we have not tested these compounds ourselves. We are currently exploring ways to measure nucleotide exchange in order to address this suggestion, but as Reviewer #4 points out these experiments are beyond the feasibility of the current study. We emphasize again that the nucleotide exchange model is the most parsimonious explanation for all of our data, and we can not imagine another less complicated way that allows hydrolysis-dead Hsp90 to progress from the ATP to ADP state.

6. How common were EA suppressors? How many fast growing colonies to total colonies? Did the authors recover any A33E direct revertants?

We acknowledge that our screen was not saturated. We used hHsp90 β -E42A as the target of the screen. Out of ~ 10,000 primary transformants (by the hHsp90 β -E42A mutagenized library), there were approximately 50 fast growing colonies on FOA. The vast majority of these fast growing colonies, all but 5, expressed wild type Hsp82 from the rescued plasmid. These false positives were mostly from inter-plasmid recombination (between the URA3 Hsp82 parental plasmid in strain MR1075 and the incoming mutagenized TRP1 hHsp90 β -E42A plasmid) or rarely from mutations in the URA3 gene of the parental plasmid that allowed it to escape FOA selection. We took advantage of the fact that cells expressing wild type hHsp90 β grow poorly at 37°C (Hsp82-expressing cells grow well at 37°C) to screen against the false positives. The five that grew well at 30°C but not 37°C were sequenced and found to have E42A and one (4 of 5) or two (1 of 5) secondary substitutions. The two individual substitutions in the most conserved residues, T179A and E384K (hHsp90 β numbering), are presented here. The others are currently the focus of ongoing study.

Our hHsp90 β -E42A allele uses the GCT codon, so in order to get reversion back to glutamate (GAA or GAG) two of the bases of this codon would have to be changed, an

extremely low probability event (A42D - GAT - would be possible with one base change but we did not observe it).

7. Line 349 “While we cannot at this time explain the mechanism underlying these observations, 350 it is obvious that EA alters the conformational response to the different nucleotides, and that 351 non-hydrolysable analogs do not mimic the effect of ATP on wild type Hsp90’s movements.” I am not sure that this is true. Is there evidence that rules out transient occupation of an ATP bound state by WT that is distinct from the AMPPNP bound state?

We regret the confusion caused by the second part of that sentence. Our aim was to emphasize that EA flipped the response to ATP vs. AMPPNP/ATP γ S and that the analogs do not have the same effect as ATP on wild type Hsp82. We have modified the sentence to: “Regardless, it is obvious that non-hydrolysable analogs and ATP have different effects on the movements of wild type Hsp90 and that EA alters the conformational response to the different nucleotides.” (revision lines 371-3)

REVIEWERS' COMMENTS

Reviewer #1 (Remarks to the Author):

The authors have addressed the comments in a rebuttal letter and revised manuscript, but my concerns remain. I reply below to the authors response to the most critical ones.

Concern:

There is a misinterpretation of PET and FRET fluorescence data of modified Hsp90 measured in response to nucleotide binding, shown in Figure 2 and analyzed and discussed on pages 6-7 and 11-13. The authors erroneously interpret the lack of amplitude (signal change) of the fluorescence decay measured for the wild-type construct in response to application of ATP, which contrasts with the large-amplitude decay measured in response to application of non-hydrolysable ATP analogues, as a lack of conformational change (Figures 2B and 3E). But the small amplitudes of the decays measured in response to ATP result from dynamic binding and hydrolysis of the nucleotide in steady state, which is possible for ATP but not for non-hydrolysable analogues. In the ATP experiment the Hsp90 molecules cycle stochastically through the closed- and open-clamp conformations, driven by nucleotide binding and hydrolysis. In this dynamic equilibrium there is always a mixture of open and closed conformations present in the large ensemble of Hsp90 molecules probed at a time, and thus the apparently small change of fluorescence signal. After the short initial phase of nucleotide binding, which is observable as a small PET decay or FRET rise, the system is in steady state where Hsp90 molecules bind and hydrolyze ATP dynamically and continuously (dynamic and continuous closing and opening of the molecular clamp), exactly as predicted from classical enzyme kinetics. The lack of amplitude in the ATP experiments shown in Figs. 2B and 2E is thus not a result of a lack of conformational change.

However, it is worth noting that the relative difference of decay amplitudes in the AMP-PNP and ATP time traces contains additional interesting information. It shows that the opening event of the Hsp90 clamp after hydrolysis is faster than the closing event after ATP-binding to the apo state. The equilibrium constant of clamp closing is $K = k_c/k_o$ (with k_c = the rate constant for closing and k_o = the rate constant for opening) and thus the small signal change measured in the steady-state dynamic equilibrium. The result is in reasonable agreement with microscopic rate constants of clamp closure and opening extracted from single molecule fluorescence intensity time traces measured in the presence of excess ATP, showing that opening is about five-fold faster than closing (ref. 8).

Author response:

1. Reviewer #1 states that the PET & FRET experiments with wild type Hsp82 and ATP do not report on the conformation of Hsp82. This argument rests solely on presumed nucleotide binding dynamics driven by ATP hydrolysis. However, in the time course of these experiments, 20 min (in the presence of nucleotide), there is little to no measurable ATP hydrolysis occurring, since, as Reviewer #1 points out

above, “Hsp90’s ATPase activity is unusually low.” Considering this fact, if ATP binding alone did indeed cause Hsp90 to form the closed-clamp conformation, then one would observe rapid quenching of fluorescence in PET or increasing FRET upon addition of ATP (as the molecules bind to the nucleotide - which is in 1000X excess - and rapidly close), followed by a slower return of fluorescence in PET or loss of FRET signal as the Hsp90 molecules start to hydrolyze ATP and reopen. A new baseline would then form once equilibrium is established. As ADP accumulated, the ATPase rate would slow even further (since ADP inhibits the hydrolysis reaction), shifting the baseline even more. But this is not what is observed. Furthermore, Hsp82-EA exhibits a very similar small PET decay/FRET increase upon addition of AMP-PNP that wild type Hsp82 does upon ATP injection (as pointed out below by Reviewer #1). This observation cannot be explained by Reviewer #1’s model of hydrolysis-driven binding dynamics, as both E33A and AMPPNP preclude hydrolysis.

Taking together the PET/FRET data and the fact that ATPase is so slow, one can argue that the formation of the closed-clamp conformation, which is required for hydrolysis to occur, is a rare event in the presence of ATP. Indeed, the small dip in amplitude in wild type/ATP PET most likely reflects the small proportion of molecules that form the closed clamp. Instead, all our data fit the most parsimonious explanation, that ATP does not induce the closed-clamp conformation like AMP-PNP or ATPgS do. E33A “flips” the conformational response to the various nucleotides, we think because of electrostatic changes to the nucleotide binding pocket that either do not allow ATP to adopt the correct conformation to induce closing or alter the ability of the protein to accurately sense the conformation of ATP.

We note that the other reviewers, #2 (“Then they investigate how these mutants affect Hsp90’s conformational dynamics”), #3 (“Using PET and FRET setups, valuable insights into nucleotide-induced conformational changes could be obtained.”) and #4 (“Using both PET and FRET approaches, they observed that EA Hsp90 formed strongly compact closed conformations...”) all appreciate our use of PET and FRET experiments to understand the

effects of E33A and/or different nucleotides on the conformational dynamics of Hsp90. Perhaps Reviewer #1 can concur that our conclusion is at least possible.

Aside from this point, Reviewer #1’s comment brings to our attention that in a single experiment in Schubert et al. (initial submission and revision reference 8), the authors used ATP in a PET experiment. We have modified the second sentence in the second section of the Results to reflect this (revision lines 180-1).

Reply:

I disagree with these arguments. In Figure 2 the authors erroneously interpret the heights of the amplitudes (relative fluorescence) in measured fluorescence intensity time traces as a criterion to judge whether the clamp closes or not. This is reasonable for an irreversible reaction induced by binding of non-hydrolysable nucleotide, which leads to a well-defined end state. But in the case of hydrolysable ATP in a common ATPase mechanism of Hsp90 sample heterogeneity arises from a dynamic equilibrium of molecules in open and closed states during ATPase-driven conformational cycling. Clamp closing induced by nucleotide binding is not fast and opening is not slow as argued by the authors above. The

opposite is the case. Hsp90 ATPase activity is rate-limited by slow closure of the clamp (Graf et al., EMBO J. 2009, 28, 602-613; refs 7 and 39 in the revised manuscript). Opening of the clamp is faster than closing (ref. 8 and 15; ref. 8 is not the only paper reporting closing and opening rates from single-molecule experiments, as stated by the authors, also ref. 15 reports them). Application of hydrolysable ATP drives Hsp90 repeatedly through its conformational cycle). In a two-state scenario of an ATPase-driven opening and closing of the clamp, the fluorescence amplitude measured in equilibrium reflects the average number of molecules in the open and in the closed states probed at a time, which is, in turn, defined by the ratio of the rate constant of closing and opening (k_c/k_o ; $k_c/k_o = K$, which is the equilibrium constant or ratio of the number of molecules in open and closed states). Thus, the baseline of low fluorescence intensity in the ATP experiment compared to the AMP-PNP experiment in Figure 2B is well explained by transient closure of the Hsp90 molecular clamp during ATPase-driven conformational cycling and does not report on a lack of closure as stated in the legend of Figure 2 and in the main text. The interpretation of the low fluorescence amplitude in the ATP experiment as lack of closure ignores the dynamic equilibrium of open and closed states in a conventional ATPase driven mechanism.

Concern:

The nucleotide exchange model depicted in Figure 7 is not reasonable. Structural studies show that in the closed-clamp conformation of Hsp90 a polypeptide segment called the “lid” is folded over the ATP binding pocket in the N-terminal domain, thus trapping the nucleotide. For nucleotide exchange to occur within the closed-clamp conformation (as depicted in Figure 7) solely the lid would need to remodel/unfold, a hypothesis that conflicts with cooperativity of conformational changes during the catalytic cycle (refs. 7 and 8).

Author response:

While it is certainly true that the lid structure is positioned over the nucleotide binding site in the AMPPNP bound (“closed-clamp” or “closed-2”) crystal structure in Ali et al., careful inspection of this structure (pdb 2cg9) in surface view reveals the presence of a hole and a channel in which the entire adenosine ring of the nucleotide can be readily observed. Residues K44, I82, E88, N92, K102 and A103 comprise the outer rim of this channel, and the bound Sba1 does not block it. In solution the diameter of the channel is likely fluctuating as the protein “breathes”. Thus, the nucleotide is not trapped completely by the lid and there is a possible “escape hatch” through which it could dissociate. We have updated the Discussion to include this possibility (revision lines 419-21) and have added Supplemental Figure 5 (new to the revision) showing the channel in the Ali et al. structure.

Reply:

It is a widely accepted view that the structural element called the lid folds over the nucleotide binding pocket and traps the nucleotide, whether it is being hydrolyzed or not. The identification of an exit

channel in structure pdb id 2CG9, which would have escaped notice in previous structural studies and allowed dissociation of nucleotide without opening of the lid as implied by the authors, is a far-fetched argument. I assume most structural biologist would object to this argument.

Reviewer #2 (Remarks to the Author):

The manuscript by Reidy et al. has been significantly improved. The conclusions are substantiated and presented more clearly. My concerns have been well addressed except for one crucial point and two minor points (see below). Altogether, this study convincingly shows that (many) Hsp90's chaperone activities do not require ATP hydrolysis in vivo. This work is very important and I support publication in Nature Communications.

Crucial point:

There cannot be any directionality in the absence of ATP hydrolysis, because then the system is in thermal equilibrium, i.e., on average in any given time interval on average the same amount of molecules transitions forward and backward. The rates might be different, but this is equalized by the state populations. This is simply the definition of equilibrium. Of course, binding of a molecule itself can induce a conformational change, but in equilibrium (i.e. in the absence of ATP hydrolysis), there will be as many binding as unbinding events on average in a given time – this is the definition of equilibrium. Therefore, there simply cannot be any directional cycle without ATP hydrolysis or any other energy source. Just binding and unbinding of molecules can never cause directionality in a cycle – this is against the law of energy conversion!

Therefore, please put arrows in both directions everywhere in the cycle of Figure 7, it is simply impossible as you draw it. Nucleotide exchange cannot cause directionality!

Please, also rephrase the following sentence (line 407), because there cannot be a start of a cycle, especially in equilibrium: “ADP can be expected to be bound to Hsp90 at the start of the cycle, as proposed [14] (Figure 7, top left).”

Minor points:

1) For the reason described above, I believe that also the following sentence in the abstract is misleading and you might rephrase it: “...exchange of ATP for ADP is critical for completion of the Hsp90 cycle.”

2) Thanks for adding the original data to Fig.2, but still several things are unclear: Is this the data from three experiments? What is AMPPNP and what ATPγS in Fig. 2B,C? The symbols are not explained. In addition, the figure caption reads: “error bars are the standard deviation” – I cannot see any error bars in this figure.

Reviewer #3 (Remarks to the Author):

The authors have answered my queries adequately in the revised version. The data added support the conclusions.

Reviewer #4 (Remarks to the Author):

The authors have addressed my concerns. From the comments of the other reviewers, it is clear that this work is controversial. In my view, the authors have done a good job of describing their results clearly and transparently. I support publication and believe this work will stimulate important follow up studies by multiple groups.

Response to Reviewers

We thank the reviewers again for their time and expertise in reviewing the second revised version of our manuscript. Changes made in the second revision are in blue (see separate manuscript file). Our responses to the reviewers' comments are in *italics* below.

Reviewer #1 (Remarks to the Author):

The authors have addressed the comments in a rebuttal letter and revised manuscript, but my concerns remain. I reply below to the authors response to the most critical ones.

Concern:

There is a misinterpretation of PET and FRET fluorescence data of modified Hsp90 measured in response to nucleotide binding, shown in Figure 2 and analyzed and discussed on pages 6-7 and 11-13. The authors erroneously interpret the lack of amplitude (signal change) of the fluorescence decay measured for the wild-type construct in response to application of ATP, which contrasts with the large-amplitude decay measured in response to application of non-hydrolysable ATP analogues, as a lack of conformational change (Figures 2B and 3E). But the small amplitudes of the decays measured in response to ATP result from dynamic binding and hydrolysis of the nucleotide in steady state, which is possible for ATP but not for non-hydrolysable analogues. In the ATP experiment the Hsp90 molecules cycle stochastically through the closed- and open-clamp conformations, driven by nucleotide binding and hydrolysis. In this dynamic equilibrium there is always a mixture of open and closed conformations present in the large ensemble of Hsp90 molecules probed at a time, and thus the apparently small change of fluorescence signal. After the short initial phase of nucleotide binding, which is observable as a small PET decay or FRET rise, the system is in steady state where Hsp90 molecules bind and hydrolyze ATP dynamically and continuously (dynamic and continuous closing and opening of the molecular clamp), exactly as predicted from classical enzyme kinetics. The lack of amplitude in the ATP experiments shown in Figs. 2B and 2E is thus not a result of a lack of conformational change.

However, it is worth noting that the relative difference of decay amplitudes in the AMP-PNP and ATP time traces contains additional interesting information. It shows that the opening event of the Hsp90 clamp after hydrolysis is faster than the closing event after ATP-binding to the apo state. The equilibrium constant of clamp closing is $K = k_c/k_o$ (with k_c = the rate constant for closing and k_o = the rate constant for opening) and thus the small signal change measured in the steady-state dynamic equilibrium. The result is in reasonable agreement with microscopic rate constants of clamp closure and opening extracted from single molecule fluorescence intensity time traces measured in the presence of excess ATP, showing that opening is about five-fold faster than closing (ref. 8).

Author response:

1. Reviewer #1 states that the PET & FRET experiments with wild type Hsp82 and ATP do not report on the conformation of Hsp82. This argument rests solely on presumed nucleotide binding dynamics driven by ATP hydrolysis. However, in the time course of these experiments, 20 min (in the presence of nucleotide), there is little to no measurable ATP hydrolysis occurring, since, as Reviewer #1 points out above, "Hsp90's ATPase activity is unusually low." Considering this fact, if ATP binding alone did indeed cause Hsp90 to form the closed-clamp conformation, then one would observe rapid quenching of fluorescence in PET or increasing FRET upon addition of ATP (as the molecules bind to the nucleotide - which is in 1000X excess - and rapidly close), followed by a slower return of fluorescence in PET or loss of FRET signal

as the Hsp90 molecules start to hydrolyze ATP and reopen. A new baseline would then form once equilibrium is established. As ADP accumulated, the ATPase rate would slow even further (since ADP inhibits the hydrolysis reaction), shifting the baseline even more. But this is not what is observed. Furthermore, Hsp82-EA exhibits a very similar small PET decay/FRET increase upon addition of AMP-PNP that wild type Hsp82 does upon ATP injection (as pointed out below by Reviewer #1). This observation cannot be explained by Reviewer #1's model of hydrolysis-driven binding dynamics, as both E33A and AMPPNP preclude hydrolysis.

Taking together the PET/FRET data and the fact that ATPase is so slow, one can argue that the formation of the closed-clamp conformation, which is required for hydrolysis to occur, is a rare event in the presence of ATP. Indeed, the small dip in amplitude in wild type/ATP PET most likely reflects the small proportion of molecules that form the closed clamp. Instead, all our data fit the most parsimonious explanation, that ATP does not induce the closed-clamp conformation like AMP-PNP or ATPgS do. E33A "flips" the conformational response to the various nucleotides, we think because of electrostatic changes to the nucleotide binding pocket that either do not allow ATP to adopt the correct conformation to induce closing or alter the ability of the protein to accurately sense the conformation of ATP.

We note that the other reviewers, #2 ("Then they investigate how these mutants affect Hsp90's conformational dynamics"), #3 ("Using PET and FRET setups, valuable insights into nucleotide-induced conformational changes could be obtained.") and #4 ("Using both PET and FRET approaches, they observed that EA Hsp90 formed strongly compact closed conformations...") all appreciate our use of PET and FRET experiments to understand the effects of E33A and/or different nucleotides on the conformational dynamics of Hsp90. Perhaps Reviewer #1 can concur that our conclusion is at least possible.

Aside from this point, Reviewer #1's comment brings to our attention that in a single experiment in Schubert et al. (initial submission and revision reference 8), the authors used ATP in a PET experiment. We have modified the second sentence in the second section of the Results to reflect this (revision lines 180-1).

Reply:

I disagree with these arguments. In Figure 2 the authors erroneously interpret the heights of the amplitudes (relative fluorescence) in measured fluorescence intensity time traces as a criterion to judge whether the clamp closes or not. This is reasonable for an irreversible reaction induced by binding of non-hydrolysable nucleotide, which leads to a well-defined end state. But in the case of hydrolysable ATP in a common ATPase mechanism of Hsp90 sample heterogeneity arises from a dynamic equilibrium of molecules in open and closed states during ATPase-driven conformational cycling. Clamp closing induced by nucleotide binding is not fast and opening is not slow as argued by the authors above. The opposite is the case. Hsp90 ATPase activity is rate-limited by slow closure of the clamp (Graf et al., EMBO J. 2009, 28, 602-613; refs 7 and 39 in the revised manuscript). Opening of the clamp is faster than closing (ref. 8 and 15; ref. 8 is not the only paper reporting closing and opening rates from single-molecule experiments, as stated by the authors, also ref. 15 reports them).

(We were specifically referring to the use of ATP in PET experiments. Mickler et al. NSMB 2009, ref 15, used FRET.)

Application of hydrolysable ATP drives Hsp90 repeatedly through its conformational cycle). In a two-state scenario of an ATPase-driven opening and closing of the clamp, the fluorescence amplitude measured in equilibrium reflects the average number of molecules in the open and in the closed states probed at a time, which is, in turn, defined by the ratio of the rate constant of closing and opening (k_c/k_o ; $k_c/k_o = K$, which is the equilibrium constant or ratio of the number of molecules in open and closed states). Thus, the baseline of low fluorescence intensity in the

ATP experiment compared to the AMP-PNP experiment in Figure 2B is well explained by transient closure of the Hsp90 molecular clamp during ATPase-driven conformational cycling and does not report on a lack of closure as stated in the legend of Figure 2 and in the main text. The interpretation of the low fluorescence amplitude in the ATP experiment as lack of closure ignores the dynamic equilibrium of open and closed states in a conventional ATPase driven mechanism.

Unfortunately, Reviewer #1 was not satisfied with our reply to this concern. Our main point is that “a dynamic equilibrium of molecules in open and closed states during ATPase-driven conformational cycling” (emphasis added by us) is not established in this system because there is very little to no measurable ATPase occurring at the timescale of these experiments. We also note that Reviewer #1’s explanation still fails to account for the observation that Hsp82-E33A does not form the closed-clamp in complex with AMPPNP or ATP γ S, but instead resembles the effect of wild type treated with ATP. This effect simply cannot be due to dynamic equilibrium caused by hydrolysis since in this case hydrolysis is doubly impossible. Finally, we regret not pointing out in our previous response that addition of ATP actually made the PET signal in lid closing increase, suggesting it shifted the equilibrium of open-lid molecules more to the open side (or constitutively activates the “burst” referred to Schulze, et al Nat Chem Bio 2016, ref 7, which is also not lid closing) as shown in Supplementary Fig 2B. This finding is very difficult to explain by Reviewer #1’s reasoning and frankly does not make a lot of sense within the current model of Hsp90 function.

These findings and our disagreements regarding their mechanism indicate that the current understanding of the role of ATP in Hsp90 function might be reevaluated.

Concern:

The nucleotide exchange model depicted in Figure 7 is not reasonable. Structural studies show that in the closed-clamp conformation of Hsp90 a polypeptide segment called the “lid” is folded over the ATP binding pocket in the N-terminal domain, thus trapping the nucleotide. For nucleotide exchange to occur within the closed-clamp conformation (as depicted in Figure 7) solely the lid would need to remodel/unfold, a hypothesis that conflicts with cooperativity of conformational changes during the catalytic cycle (refs. 7 and 8).

Author response:

While it is certainly true that the lid structure is positioned over the nucleotide binding site in the AMPPNP bound (“closed-clamp” or “closed-2”) crystal structure in Ali et al., careful inspection of this structure (pdb 2cg9) in surface view reveals the presence of a hole and a channel in which the entire adenosine ring of the nucleotide can be readily observed. Residues K44, I82, E88, N92, K102 and A103 comprise the outer rim of this channel, and the bound Sba1 does not block it. In solution the diameter of the channel is likely fluctuating as the protein “breathes”. Thus, the nucleotide is not trapped completely by the lid and there is a possible “escape hatch” through which it could dissociate. We have updated the Discussion to include this possibility (revision lines 419-21) and have added Supplemental Figure 5 (new to the revision) showing the channel in the Ali et al. structure.

Reply:

It is a widely accepted view that the structural element called the lid folds over the nucleotide binding pocket and traps the nucleotide, whether it is being hydrolyzed or not. The identification of an exit channel in structure pdb id 2CG9, which would have escaped notice in previous structural studies and allowed dissociation of nucleotide without opening of the lid as implied by the authors, is a far-fetched argument. I assume most structural biologist would object to this

argument.

See our response to Reviewer #2 below regarding changes to Figure 7 in the second revision and their underlying rationale. We have amended Supplementary Figure 5 to include an image showing the analogous channel in the Verba et al. human Hsp90 β -Cdc37-Cdk4 structure (pdb id 5fwk, ref 51), strongly suggesting this channel is a feature, not a bug, of Hsp90. Regardless, we made no implication as to what escaped the notice of structural biologists. As we pointed out in our reply to the previous concern, the lid does not fold over the nucleotide binding pocket if only ATP is introduced, but it does if AMPPNP is (Supplementary Fig. 2B). This experimental observation is at odds with the “widely accepted view” yet totally in line with our previous demonstration that Hsp82 closed clamps formed in the presence of AMPPNP are opened simply by the addition of ADP (Reidy & Masison JMB 2020; ref 33). If the lid “trapped” the nucleotide then this observation would be impossible. We have added this point to the Discussion in the second revision (lines 422-4).

Reviewer #2 (Remarks to the Author):

The manuscript by Reidy et al. has been significantly improved. The conclusions are substantiated and presented more clearly. My concerns have been well addressed except for one crucial point and two minor points (see below). Altogether, this study convincingly shows that (many) Hsp90's chaperone activities do not require ATP hydrolysis in vivo. This work is very important and I support publication in Nature Communications.

We thank Reviewer #2 for the kind words and support of our work.

Crucial point:

There cannot be any directionality in the absence of ATP hydrolysis, because then the system is in thermal equilibrium, i.e., on average in any given time interval on average the same amount of molecules transitions forward and backward. The rates might be different, but this is equalized by the state populations. This is simply the definition of equilibrium. Of course, binding of a molecule itself can induce a conformational change, but in equilibrium (i.e. in the absence of ATP hydrolysis), there will be as many binding as unbinding events on average in a given time – this is the definition of equilibrium. Therefore, there simply cannot be any directional cycle without ATP hydrolysis or any other energy source. Just binding and unbinding of molecules can never cause directionality in a cycle – this is against the law of energy conversion! Therefore, please put arrows in both directions everywhere in the cycle of Figure 7, it is simply impossible as you draw it. Nucleotide exchange cannot cause directionality!

We have redrawn Figure 7 to include a depiction of the client bound to the closed clamp and to indicate the binding and unbinding of co-chaperones (individual co-chaperones are not depicted). We have also made slight modifications to the Discussion pertaining to Figure 7 (lines 403-19). Binding of the client and the associated actions of co-chaperones drives formation of the closed clamp, as the Agard group showed via cryo-EM of complexes isolated from cells (pdb id 5fwk; Verba et al. Science 2016; ref 51). We now cite this finding in the second revision (line 412). Once formed, the closed clamp can reopen if the ATP is hydrolyzed or exchanged for ADP. This must be the case since hydrolysis defective Hsp90 can function in vivo! If “directionality” with respect to Hsp90 is going from open to closed to open again, then nucleotide exchange can cause it, as we showed previously that Hsp90 can go from closed-clamp (AMPPNP) to open solely by the addition of ADP (Reidy & Masison JMB 2020, ref 33) – no hydrolysis was occurring in those experiments. Reference to this finding has been added to

the modified Discussion (lines 422-4). Finally, enthalpic effects due to the binding and release of co-chaperones and the client almost certainly play a directional role in the cell, which in the end has the final say on whether a particular activity of Hsp90 is required for it to function.

Please, also rephrase the following sentence (line 407), because there cannot be a start of a cycle, especially in equilibrium: "ADP can be expected to be bound to Hsp90 at the start of the cycle, as proposed [14] (Figure 7, top left)."

We rephrased the sentence to "...ADP can be expected to be bound to Hsp90 after hydrolysis, as proposed..." (lines 407-8)

Minor points:

1) For the reason described above, I believe that also the following sentence in the abstract is misleading and you might rephrase it: "...exchange of ATP for ADP is critical for completion of the Hsp90 cycle."

We rephrased the sentence to "...exchange of ATP for ADP is critical for Hsp90 function." (line 28)

2) Thanks for adding the original data to Fig.2, but still several things are unclear: Is this the data from three experiments? What is AMPPNP and what ATPγS in Fig. 2B,C? The symbols are not explained. In addition, the figure caption reads: "error bars are the standard deviation" – I cannot see any error bars in this figure.

We apologize for the confusion caused by our unclear figure legend. We have updated the legend of Figure 2 in the second revision to address all the points in your concern.

Reviewer #3 (Remarks to the Author):

The authors have answered my queries adequately in the revised version. The data added support the conclusions.

We thank Reviewer #3 for their time and expertise in reviewing our work.

Reviewer #4 (Remarks to the Author):

The authors have addressed my concerns. From the comments of the other reviewers, it is clear that this work is controversial. In my view, the authors have done a good job of describing their results clearly and transparently. I support publication and believe this work will stimulate important follow up studies by multiple groups.

We thank Reviewer #4 for the kind words and support of our work, which we would perhaps prefer to characterize as more thought-provoking than controversial.